# RESIDUAL CONNECTIONS AND NORMALIZATION CAN PROVABLY PREVENT OVERSMOOTHING IN GNNS

**Michael Scholkemper**[*]
Department of Computer Science
RWTH Aachen University
`scholkemper@cs.rwth-aachen.de`

**Xinyi Wu**[*]
Institute for Data, Systems, and Society
Massachusetts Institute of Technology
`xinyiwu@mit.edu`

**Ali Jadbabaie**
Institute for Data, Systems, and Society
Massachusetts Institute of Technology
`jadbabai@mit.edu`

**Michael T. Schaub**
Department of Computer Science
RWTH Aachen University
`schaub@cs.rwth-aachen.de`

## ABSTRACT

Residual connections and normalization layers have become standard design choices for graph neural networks (GNNs), and were proposed as solutions to the mitigate the oversmoothing problem in GNNs. However, how exactly these methods help alleviate the oversmoothing problem from a theoretical perspective is not well understood. In this work, we provide a formal and precise characterization of (linearized) GNNs with residual connections and normalization layers. We establish that (a) for residual connections, the incorporation of the initial features at each layer can prevent the signal from becoming too smooth, and determines the subspace of possible node representations; (b) batch normalization prevents a complete collapse of the output embedding space to a one-dimensional subspace through the individual rescaling of each column of the feature matrix. This results in the convergence of node representations to the top-$k$ eigenspace of the message-passing operator; (c) moreover, we show that the centering step of a normalization layer — which can be understood as a projection — alters the graph signal in message-passing in such a way that relevant information can become harder to extract. Building on the last theoretical insight, we introduce GraphNormv2, a novel and principled normalization layer. GraphNormv2 features a learnable centering step designed to preserve the integrity of the original graph signal. Experimental results corroborate the effectiveness of our method, demonstrating improved performance across various GNN architectures and tasks.

## 1 INTRODUCTION

In recent years, graph neural networks (GNNs) have gained significant popularity due to their ability to process complex graph-structured data and extract features in an end-to-end trainable fashion (Gori et al., 2005; Scarselli et al., 2009; Bruna et al., 2014; Duvenaud et al., 2015; Defferrard et al., 2016; Kipf & Welling, 2017; Veličković et al., 2018). They have shown empirical success in a highly diverse set of problems (Battaglia et al., 2016; Wu et al., 2020; Xu et al., 2019; Morris et al., 2019; Jha et al., 2022; Cappart et al., 2021). Most GNNs follow a message-passing paradigm (Gilmer et al., 2017), where node representations are learned by recursively aggregating and transforming the representations of the neighboring nodes. Through repeated message-passing on the graph, the graph information is implicitly incorporated.

A prevalent problem in message-passing GNNs is their tendency to oversmooth (Oono & Suzuki, 2020; Cai et al., 2021; Wu et al., 2023a; Keriven, 2022), which refers to the observation that node signals (or representations) tend to contract to a one-dimensional subspace as the number of layers increases. While a certain amount of smoothing is desirable to average out noise in the node features

---

[*]Both authors contributed equally.

and render information extraction from the graph more reliable (Keriven, 2022; Wu et al., 2023b), excessive smoothing leads to an information loss as node signals become virtually indistinguishable and can thus not be exploited for downstream tasks. For this reason, most GNN architectures use only few message-passing layers (Kipf & Welling, 2017; Veličković et al., 2018; Xu et al., 2019). This contrasts with the trend in deep learning, where much deeper architectures are considered preferable (He et al., 2016; Kaplan et al., 2020)

To mitigate the oversmoothing problem for GNNs, several practical solutions have been proposed. In particular, both residual connections (Chen et al., 2020; Klicpera et al., 2019) and normalization layers (Zhao & Akoglu, 2020; Zhou et al., 2021; Guo et al., 2023) have been specifically proposed to address the oversmoothing problem. Despite empirical observations that the introduction of these methods can alleviate oversmoothing and enable deeper architectures, a theoretical analysis on the effects of these methods on oversmoothing and the resultant expressive power of GNNs is still lacking. More technically, the analysis of oversmoothing typically relies on a single node similarity measure as a description of the whole underlying system (Oono & Suzuki, 2020; Cai & Wang, 2020; Wu et al., 2023a). Such measures, however, can only identify a complete collapse of the node signals to one-dimensional subspace and are unable to capture the more refined geometry of the system beyond a complete collapse. These observations motivate the following questions:

> *How do residual connections and normalization layers affect oversmoothing and therefore the practical expressive power of GNNs? How do they compare?*

In this work, we answer the above questions with a refined characterization of the underlying system of linearized GNNs (Keriven, 2022; Wu et al., 2023b) with residual connections and normalization layers. In particular, we establish that both methods can alleviate oversmoothing by preventing node signals from a complete collapse to a one-dimensional subspace, contrary to the case for standard GNNs (Oono & Suzuki, 2020; Cai & Wang, 2020; Wu et al., 2023a).

**Our contributions can be summarized as follows:**

- We characterize the system of linearized GNNs with residual connections. We show that residual connections prevent complete rank collapse to a one-dimensional subspace by incorporating the initial features at each step. As a result, the subspace of possible node representations that a GNN can compute is determined by the initial features.
- We analyze the system of linearized GNNs with normalization layers and show that normalization layers prevent oversmoothing of node signals through the scaling step. Nonetheless, the node representations exponentially converge to the top-$k$ eigenspace of the message-passing operator.
- We separately identify the role of the centering step in normalization layers. Our results suggest that the centering step can distort the message-passing in undesired ways. Consequently, relevant graph information becomes dampened and can even be lost in message-passing.
- Based on these theoretical findings, we propose a novel normalization layer named GraphNormv2 that learns a centering operation that does not distort the subspaces of the message-passing operator in an uncontrolled way, but contains a learnable projection. Experimental results show the effectiveness of our proposed method.

## 2 RELATED WORK

**Oversmoothing in GNNs**   Oversmoothing is a known challenge for developing deeper and more powerful GNNs, and many techniques have been proposed to mitigate the issue in practice. Among them, residual connections (Chen et al., 2020) and normalization layers (Zhao & Akoglu, 2020; Zhou et al., 2021; Guo et al., 2023) are popular methods that have empirically been shown to mitigate oversmoothing to certain extent. While many theoretical works on the underlying mechanism of oversmoothing exist (Oono & Suzuki, 2020; Cai et al., 2021; Keriven, 2022; Wu et al., 2023b;a), these studies focus on standard GNNs without these additional modules. It is thus still an open question how residual connections and normalization can mitigate oversmoothing and subsequently affect the practical expressive power of GNNs from a theoretical perspective.

**Theoretical studies on residual connections and normalization in deep learning**   The empirical success of residual connections and normalization in enhancing training deep neural networks has inspired research into their underlying mechanisms (Arora et al., 2018; Daneshmand et al.,

2021; Yang et al., 2019; Liu et al., 2019; Hardt & Ma, 2017; Esteve et al., 2020). Specifically, it has been shown that batch normalization avoids rank collapse for randomly initialized deep linear networks (Daneshmand et al., 2020) and that residual connections alleviate rank collapse in transformers (Dong et al., 2021). Rank collapse of node representations due to oversmoothing has also been a notable issue in building deeper GNNs. However, a theoretical analysis of how residual connections and normalization layers combat oversmoothing and their additional effects on message-passing in GNNs is still lacking.

## 3 NOTATION AND PRELIMINARIES

**Notation**  We use the shorthand $[n] := \{1, \ldots, n\}$. We denote the zero-vector by $\mathbf{0}$, the all-one vector by $\mathbf{1}$ and the identity matrix by $I$. Let $\|\cdot\|_2$, $\|\cdot\|_F$ be the 2-norm and Frobenius norm, respectively. Lastly, for a matrix $M$, we denote its $i^{th}$ row by $M_{i,:}$ and $j^{th}$ column by $M_{:,j}$.

**Graph Neural Networks**  Most message-passing GNN models — which we will simply refer to as GNNs from now on — can be described by the following update equation:

$$X^{(t+1)} = \sigma(AX^{(t)}W^{(t)}), \tag{1}$$

where $X^{(t)}, X^{(t+1)} \in \mathbb{R}^{n \times k}$ are the input and output node representations of the $t^{\text{th}}$ layer, respectively; $\sigma(\cdot)$ is an element-wise non-linearity such as the ReLU function; $W^{(t)} \in \mathbb{R}^{k \times k}$ is a learnable linear transformation and $A \in \mathbb{R}_{\geq 0}^{n \times n}$ represents a message-passing operation and reflects the graph structure. For example, if $A$ is the graph adjacency matrix, $A = A_{\text{adj}}$ we recover the Graph Isomorphism Network (GIN) (Xu et al., 2019), and for $A = D^{-\frac{1}{2}} A_{\text{adj}} D^{-\frac{1}{2}}$, where $D = \text{diag}(A_{\text{adj}}\mathbf{1})$ is the diagonal degree matrix, we obtain a Graph Convolutional Network (GCN) (Kipf & Welling, 2017).

Throughout the paper, we assume $A$ to be symmetric and non-negative. Furthermore, we assume that eigenvalues $\lambda_i$ of a matrix are organized in a non-increasing order in terms of absolute value: $|\lambda_1| \geq |\lambda_2| \geq \cdots \geq |\lambda_n|$. For simplicity, we assume the columns of the initial features $X^{(0)}$ to be normalized, i.e. $\|X_{:,i}^{(0)}\|_2 = 1$.

**Normalization Layers**  Normalization has been believed to be beneficial to deep neural networks for nearly two decades (LeCun et al., 2002). Most normalization layers perform a *centering* operation and a *scaling* operation on the input features. Centering usually consists of subtracting the mean, centering the features around zero along the chosen dimension. Scaling usually consists of scaling the features such that the features along the chosen dimension have unit variance. The two most popular approaches are *batch* and *layer* normalization (Ioffe & Szegedy, 2015; Ba et al., 2016). In our work, we focus on batch normalization (BatchNorm), denoted as $\text{BN}(\cdot)$, which is defined as follows: Let $X \in \mathbb{R}^{n \times k}$, then

$$\text{BN}(X) = [\cdots, \text{bn}(X_{:,i}), \cdots],$$

where $\text{bn}(x) = (x - \sum_{i=1}^{n} x_i/n)/\|x - \sum_{i=1}^{n} x_i/n\|_2$. We note that in the case of GNNs, especially graph classification tasks, batches may contain nodes from different graphs. In our analysis, we consider only the nodes in the same graph for normalization and nodes in different graphs do not influence each other. This is sometimes called *instance* normalization (Ulyanov et al., 2016).

**GraphNorm**  In Cai et al. (2021), the authors proposed GraphNorm, a normalization layer specifically designed for GNNs that is like BatchNorm in terms of acting on the columns of the feature matrix. However, compared to BatchNorm, instead of subtracting the mean, the method learns to subtract an $\tau_j$ portion of the mean for the $j^{th}$ column:

$$\text{GraphNorm}(X_{:,j}) = \gamma_j \frac{X_{:,j} - \tau_j \mathbf{1}\mathbf{1}^\top X_{:,j}/n}{\sigma_j} + \beta_j,$$

where $\sigma_j = \|X_{:,j} - \tau_j \mathbf{1}\mathbf{1}^\top X_{:,j}/n\|_2/\sqrt{n}$, $\gamma_j, \beta_j$ are the learnable affine parameters as in the implementation of other normalization methods (Ba et al., 2016; Ioffe & Szegedy, 2015). Notably, by introducing $\tau_j$ for each feature dimension $j$, (Cai et al., 2021) claims an advantage of GraphNorm over the original batch normalization in that GraphNorm is able to learn how much of the information to keep in the mean rather than always subtracting the entire mean away.

**Weisfeiler-Leman and Structural Eigenvectors**   In theoretical studies about GNNs, an algorithm that comes up frequently is the so-called Weisfeiler-Leman (WL) algorithm (Weisfeiler & Leman, 1968). This algorithm iteratively assigns a color $c(v) \in \mathbb{N}$ to each node $v$ starting from a constant initial coloring $c^{(0)}(v) = 1$ for all nodes. In each iteration, an update of the following form is computed: $c^{(t+1)}(v) = \text{hash}\left(c^{(t)}(v), \{\{c^{(t)}(x) \mid x \in \mathcal{N}(v)\}\}\right)$, where hash is an injective hash function, $\{\{\cdot\}\}$ denotes a multiset in which elements can appear more than once, and $\mathcal{N}(v)$ is the set of neighboring nodes of $v$. The algorithm returns the final colors $c^{(\infty)}$ when the partition $\{(c^{(t)})^{-1}(c^{(t)}(v))|v \in V(G)\}$ no longer changes for consecutive update $t$. For GNNs, Morris et al. (2019) and Xu et al. (2019) showed that for graphs with constant initial node features, GNNs cannot compute different features for nodes that are in the same class in the final coloring $c^{(\infty)}$.

For this paper, an equivalent algebraic perspective of the WL algorithm[1] will be more useful (see Appendix A.1 for a detailed discussion): Given $c^{(\infty)}$ with $\text{Im}(c^{(\infty)}) = \{c_1, .., c_k\}$, define $H \in \{0,1\}^{n \times k}$ such that $H_{v,i} = 1$ if and only if $c^{(\infty)}(v) = c_i$. It holds that

$$AH = HA^\pi, \tag{2}$$

where $A^\pi \in \mathbb{R}^{k \times k}$ is the adjacency matrix of the *quotient graph*, which is fixed given $A$ and $H$. In words, a node in the quotient graph represents a class of nodes in the original graph who share the same number of neighbors in each final color.

Most relevantly, the adjacency matrix $A$ of the original graph inherits all the eigenpairs from the quotient graph: If $\nu^\pi$ is the eigenvector of $A^\pi$ with eigenvalue $\lambda$, then $H\nu^\pi$ is an eigenvector of $A$ with eigenvalue $\lambda$. We call such eigenvectors of $A$ the *structural eigenvectors*. These eigenvectors are important for understanding dynamical systems on graphs (Schaub et al., 2016; Yuan et al., 2013), and play a role for centrality measures such as PageRank (Sánchez-García, 2020) and others (Stamm et al., 2023). In fact, from the results of Morris et al. (2019) and Xu et al. (2019), we can directly infer that given constant initial features, GNNs effectively compute node features on this quotient graph, meaning the features lie in the space spanned by the structural eigenvectors.

## 4   MAIN RESULTS: DEFYING OVERSMOOTHING

Both residual connections and normalization techniques are widely recognized as effective methods for mitigating oversmoothing in GNNs (Chen et al., 2020; Klicpera et al., 2019; Zhou et al., 2021; Zhao & Akoglu, 2020; Guo et al., 2023). In this section, we provide theoretical evidence that these approaches not only mitigate but *provably* prevent oversmoothing. Previous studies have demonstrated that oversmoothing occurs exponentially in standard Graph Convolutional Networks (GCNs) (Oono & Suzuki, 2020; Cai et al., 2021) and similarly in random walk GCNs and more general attention-based GNNs (Wu et al., 2023a). These findings suggest that for GNNs with non-diverging weights, repeated message-passing invariably leads to the collapse of node signals into a one-dimensional subspace, regardless of initial features. However, our analysis reveals that these results do not hold for GNNs employing residual connections or BatchNorm. Furthermore, we offer a precise characterization of the convergence space for GNNs utilizing these techniques, providing deeper insights into their effects.

For the following theoretical analysis, we investigate a linearized GNN, meaning that $\sigma(\cdot)$ is the identity map. For simplicity, we assume, if not specified otherwise, that the weights $(W_1^{(t)})_{i,j}, (W_2^{(t)})_{i,j}, W_{i,j}^{(t)}$ are randomly sampled i.i.d. Gaussians, which is typical for GNNs before training. Such a setting is relevant in practice as oversmoothing is present in GNNs before training has started, making the gradients used for back propagation almost vanish and training of the network becomes much harder. Yet, most results hold under more general conditions. The complete proofs of all the results in the main text and the results under general conditions are provided in Appendix B.

To define oversmoothing, a common approach is to analyze a function $\mu(\cdot)$, which measures how far the node features $X^{(t)}$ are away from collapsing to a one-dimensional subspace. One then shows that $\mu(X^{(t)}) \to 0$ as $t \to \infty$, or – even stronger – that this convergence happens at an exponential rate, i.e. $\mu(X^{(t)}) \le C_1 e^{-C_2 t}$ for some $C_1, C_2 > 0$. There are multiple such measures of oversmoothing

---

[1]This algebraic characterisation is also known as the *coarsest equitable partition (cEP)* of the graph.

in previous works. For example, (Cai & Wang, 2020) use the graph Dirichlet energy $\mathcal{E}(X)$, whereas (Rusch et al., 2023; Wu et al., 2023a) use the squared difference to the mean over each row and (Oono & Suzuki, 2020) use the distance to the dominant eigenspace of the message-passing operator $A$. To allow for an analysis that takes all these perspectives into account, we will focus our analysis on the distance to a general one-dimensional subspace spanned by a unit vector $v \in \mathbb{R}^n$:

$$\mu_v(X) := \|X - vv^\top X\|_F^2 = \|(I - vv^\top)X\|_F^2. \tag{3}$$

Thus, when the graph is ergodic and we choose $v$ to be the dominant eigenvector, we recover the measure used in (Oono & Suzuki, 2020). With $v = \mathbf{1}/\sqrt{n}$, we recover the measure used in (Rusch et al., 2023; Wu et al., 2023a) and for $v = d/\|d\|_2$ where $d = D^{\frac{1}{2}}\mathbf{1}$, we recover an equivalent measure to the graph Dirichlet energy $\mathcal{E}(X)$ used in (Cai & Wang, 2020). Here equivalence means that there exist constants $C_1, C_2 > 0$ such that

$$C_1\mu_{d/\|d\|_2}(X) \leq \mathcal{E}(X) \leq C_2\mu_{d/\|d\|_2}(X).$$

This allows us to translate statements about oversmoothing in terms of $\mu_v$ to any of the other measures mentioned. A detailed discussion can be found in Appendix A.2.

## 4.1 RESIDUAL CONNECTIONS PREVENT COMPLETE COLLAPSE

For our analysis of residual connections, we focus on the commonly used initial residual connections, which are deployed in architectures like GCNII (Chen et al., 2020). Such residual connections are closely related to the Personalized PageRank propagation (Klicpera et al., 2019). For generality, we write the unified layer-wise update rule for both methods as follows:

$$X^{(t+1)} = (1 - \alpha)AX^{(t)}W_1^{(t)} + \alpha X^{(0)}W_2^{(t)}, \tag{4}$$

where $\alpha X^{(0)}$ corresponds to the initial residual connection, and $\alpha \in (0, 1)$ can be seen as the strength of residual connections or alternatively as the teleportation probability in the Personalized PageRank Propagation. Note that for the Personalized PageRank Propagation method (APPNP) proposed in (Klicpera et al., 2019), $A = D^{-\frac{1}{2}}A_{\text{adj}}D^{-\frac{1}{2}}$, $W_1^{(t)} = I_k$ for all $t \geq 0$.

Intuitively, compared to the case for standard message-passing GNNs in (1), at each update step, a linear combination of the residual signal $\alpha X^{(0)}$ is now added to the features. As long as the weight matrices are not chosen such that they annihilate the residual signal, this will prevent the features from collapsing to a smaller subspace. This implies that $\mu_v(X^{(t)})$ would be strictly greater than zero. In fact, $\mu_v(X^{(t)})$ can even be bounded away from 0, showing that oversmoothing does not happen.

**Proposition 4.1.** *Let $v \in \mathbb{R}$ s.t. $\|v\|_2 = 1$. If $\mu_v(X^{(0)}) > 0$, then w.h.p $\exists c > 0$ s.t. $\mu_v(X^{(t)}) \geq c$.*

The above result suggests that, with proper initialization of node features, initial residual connections will alleviate oversmoothing with high probability, meaning the features will not be smooth after iterative message-passing at initialization. Nonetheless, it is worth noting that the node similarity measure $\mu_v(X)$ can only identify a complete collapse to a one-dimensional subspace spanned by $v$, in which case $\mu_v(X)$ equals zero. Even if $\mu_v(X)$ can remain strictly positive with residual connections, this does not eliminate the possibility that there may still be partial collapse of the signal to a lower-dimensional subspace. However, as we will show with the following result, with residual connections defined in (4), no such partial collapse occurs.

**Proposition 4.2.** *Let $x_i = X_{:,i}^{(0)}$, $\|x_i\|_2 = 1$ and let each $(W_l^{(t)})_{y,z} \overset{\text{i.i.d.}}{\sim} \mathcal{N}(\eta, s^2)$. Then for any $\epsilon > 0$, $\|x_i^\top X^{(t)}\|_2 \geq \epsilon$ with probability at least $p = 1 - \exp\left(-\frac{\epsilon^2}{2\alpha^2 s^2}\right)$.*

This result essentially says that a part of the initial signal is maintained after each layer with high probability. We can further give a more refined characterization of what node features the system in (4) can compute after repeated rounds of message-passing.

**Proposition 4.3.** *Let $\text{Kr}(A, X^{(0)}) = \text{Span}(\{A^{i-1}X_{:,j}^{(0)}\}_{i \in [n], j \in [k]})$ be the Krylov subspace. Let $Y \in \mathbb{R}^{n \times k}$. Then there exist a $T \in \mathbb{N}$ and sequence of weights $W_1^{(0)}, W_2^{(0)}, \cdots, W_1^{(T-1)}, W_2^{(T-1)}$ such that $X^{(T)} = Y$ if and only if $Y \in \text{Kr}(A, X^{(0)})$.*

The result shows that for a message-passing GNN with residual connections, the subspace that the embedding $X^{(t)}$ lies in is governed by the initial features $X^{(0)}$ together with the message-passing operator $A$ and that any such signal is reachable by a sequence of weights. This is in contrast with the behavior of standard message-passing GNNs, for which node representations eventually becomes "memoryless", i.e., independent of initial features. In particular, for standard GCNs, the subspace the system converges to is completely governed by the message-passing operator $A$ (Oono & Suzuki, 2020).

**Remark 4.4.** Proposition 4.3 connects closely to Theorem 2 in Chen et al. (2020). Specifically, Theorem 2 in Chen et al. (2020) implies that for any features in the Krylov subspace, there exists weights such that the corresponding GNN outputs such features. Here, we prove in addition an upper bound, that the GNN cannot express any features that lie beyond the Krylov subspace.

**Remark 4.5.** Our results also suggest that whether initial residual connections would work for oversmoothing heavily depends on the initialization of features. If chosen poorly, they may not be able to prevent oversmoothing. In particular, a constant initialization of features is unhelpful to prevent oversmoothing with initial residual connections.

## 4.2 BATCHNORM PREVENTS COMPLETE COLLAPSE

Having discussed the effect of residual connections, in this section, we switch gears and analyze how BatchNorm affects GNNs. We consider the following combination of GNNs with BatchNorm:

$$X^{(t+1)} = \text{BN}(Y^{(t+1)}), \quad Y^{(t+1)} = \sigma(A X^{(t)} W^{(t)}). \tag{5}$$

Similar to the analysis on GNNs with initial residual connections, we will show that BatchNorm prevents complete collapse of the output embedding space to a one-dimensional subspace. We again consider the case where $\sigma(\cdot)$ is the identity map. Note that even though the GNN computation is now linearized, the overall system remains non-linear because of the BatchNorm operation. Let us consider now $V_{\neq 0} \in \mathbb{R}^{n \times k'}$ to be the matrix of eigenvectors associated to non-zero eigenvalues of $(I_n - \mathbf{1}\mathbf{1}^\top / n)A$.

**Proposition 4.6.** Let $v \in \mathbb{R}^n$ s.t. $\|v\|_2 = 1$ and $v^T \mathbf{1} > 0$. Let $\text{Rank}(V_{\neq 0}^\top X^{(0)}) > 1$, then there exists $c(v) > 0$ such that $\mu_v(X^{(t)}) \geq c(v)$ for all $t \geq 1$ with probability 1 .

The above result suggests that with BatchNorm, $\mu_v(X^{(t)})$ is maintained strictly greater than a nontrivial constant at each layer, indicating that complete collapse to a one-dimensional subspace does not happen.

**Remark 4.7.** A more general version of the above result holds for both linear and non-linear GNNs, when assuming $\sigma(\cdot)$ is injective: for $v \in \mathbb{R}^n$ such that $v^\top \mathbf{1} \neq 0$, under some regularity conditions, there exists $c(v) > 0$ such that $\mu_v(X^{(t)}) = c(v)\sqrt{k}$ for all $t \geq 1$, where $k$ is the hidden dimension of features. In particular, this would account for the case where $A$ is the adjacency matrix of the graph or the symmetric random walk matrix $D^{-1/2}A_{\text{adj}}D^{-1/2}$, and $v$ is their corresponding dominant eigenvector. See Appendix B.7 for a detailed discussion.

However, as what we have discussed for the case of residual connections, the measure $\mu_v(X)$ can only capture complete collapse to a one-dimensional subspace. In what follows, we will provide a more precise characterization of the convergence behaviors of GNNs with BatchNorm. Notably, since the scaling operation of BatchNorm guarantees that the system will not diverge or collapse, we can give an exact asymptotic characterization. Let $V_k \in \mathbb{R}^{n \times k}$ be the matrix of the top-$k$ eigenvectors of $(I_n - \mathbf{1}\mathbf{1}^\top / n)A$. We will show that the resulting linearized GNN converges to a rank-$k$ subspace spanned by the top-$k$ eigenvectors of $(I_n - \mathbf{1}\mathbf{1}^\top / n)A$.

**Proposition 4.8.** Suppose $V_k^\top X^{(0)}$ has rank $k$, then for all weights $W^{(t)}$, the GNN with BatchNorm given in (5) exponentially converges to the column space of $V_k$.

It is easy to see that the centering operation in BatchNorm is the reason why we use the *centered* message-passing operator $(I_n - \mathbf{1}\mathbf{1}^\top / n)A$ and its eigenvectors here. If we inspected BatchNorm without scaling, however, we would arrive at a completely linear system, which must converge to the dominant eigenvector (given mild assumptions). This implies that the *column-wise scaling operation is responsible for the preservation of the rank of the features.* Furthermore, there are no requirements for the weights as in the case without BatchNorm (Oono & Suzuki, 2020; Wu et al., 2023a): even

extremely large or random weights can be chosen, as the scaling ensures that the system neither diverges nor collapses.

The convergence of the linearized system in (5) to a $k$-dimensional subspace can be shown to be tight: we can choose weights such that the top-$k$ eigenvectors are exactly recovered. This is of course only possible if these eigenvectors have a nonzero eigenvalue, which we assume for the result below, in that, none of the top-$k$ eigenvectors of $(I_n - \mathbf{1}\mathbf{1}^\top/n)A$ have eigenvalue zero.

**Proposition 4.9.** *Suppose $|\hat{\lambda}_k| > 0$ and $V_k^\top X^{(0)}$ has rank $k$. For any $\epsilon > 0$, there exists $T > 0$ and a sequence of weights $W^{(0)}, W^{(1)}, ..., W^{(T)}$ such that for all $t \geq T$ and $i \in [k]$,*

$$\left\| \nu_i^\top X_{:,i}^{(t)} \right\|_2 \geq 1/\sqrt{1+\epsilon},$$

*where $\nu_i$ denotes the $i$-th eigenvector of $(I_n - \mathbf{1}\mathbf{1}^\top/n)A$.*

The above result ties in nicely with recent results showing that GNNs are strengthened through *positional encodings* (Dwivedi et al., 2021; 2023), where the features are augmented by the top-$k$ eigenvectors of the graph. This can, in some sense, be seen as emulating a deep GNN, which would converge to the top-$k$ eigenspace using BatchNorm. Yet it is worth noting that while BatchNorm improves the practical expressive power of GNNs by converging to a larger subspace, under the type of convergence given in Proposition 4.8, the information in the eigenvectors associated with small magnitude eigenvalues is still dampened after repeated message-passing. In the limit $t \to \infty$, the computation is thus still memoryless.

**Remark 4.10.** If no initial node features $X^{(0)}$ are given by the dataset, common choices in practice are to initialize the features randomly or identically for each node. In the former case, the prerequisites of 4.8 and Proposition 4.9 are satisfied. In the latter case, the conditions are not met, as $X^{(0)}$ has rank one. In that case, the system still converges. In fact, it retains its rank and converges to the dominant eigenvector of $(I_n - \mathbf{1}\mathbf{1}^\top/n)A$.

**Comparison between normalization and residual connections.** From previous sections, we have seen that both residual connections and batch normalization are able to prevent a complete collapse of the node embeddings to a one-dimensional subspace. In both cases, the embeddings converge to a larger subspace and thus oversmoothing is alleviated. However, it is clear that different mechanisms are at play to mitigate oversmoothing. With residual connections, the system is able to keep the dimensions of the initial input features by incorporating the initial features $X^{(0)}$ at each layer; while with normalization, the system converges to the subspace of the top eigenvectors of the message-passing operator through the scaling step.

## 5 CENTERING DISTORTS THE GRAPH

So far, we have analyzed the effects of residual connections and normalization layers on oversmoothing. Specifically, we have shown that the incorporation of initial features of residual connections and the scaling effect of normalization help alleviate complete rank collapse of node features. However, there are two steps in normalization layers: centering and scaling. If already scaling helps preventing a complete collapse, a natural question is *what is the role of centering in the process?* In this section, we will show that the current centering operation used in normalization layers can in fact have an undesirable effect altering the graph signal that message-passing can extract, as if message-passing happens on a different graph.

### 5.1 CENTERING INTERFERES THE STRUCTURAL EIGENVECTORS

The centering operation in normalization layers takes away the (scaled) mean across all rows in each column, and thus can be written as applying the operator $I_n - \tau\mathbf{1}\mathbf{1}^\top/n$ to the input, where $\tau$ indicates how much mean is taken away in centering. Specifically, for $\tau = 1$ we recover BatchNorm, whereas for $\tau \in \mathbb{R}$, we recover GraphNorm (Cai et al., 2021). To analyze how this step would alter the graph signal message-pasing can extract, we make use of the concepts of quotient graph and structural eigenvectors as introduced in Section 3.

Given a symmetric, non-negative adjacency matrix $A \in \mathbb{R}_{\geq 0}^{n \times n}$, let $H \in \{0, 1\}^{n \times m}$ be the indicator matrix of its final WL coloring $c^{(\infty)}$. Consider the eigenpairs $\mathcal{V} = \{..., (\lambda, \nu), ...\}$ of $A$ and divide

them into the set of structural eigenpairs $\mathcal{V}_{\text{struc}} = \{(\lambda, \nu) \in \mathcal{V} \mid \nu = H\nu^\pi\}$, and the remaining eigenpairs $\mathcal{V}_{\text{rest}} = \mathcal{V} \backslash \mathcal{V}_{\text{struc}}$. Similarly, let $\hat{\mathcal{V}} = \{..., (\hat{\lambda}, \hat{\nu}), ...\}$ be the eigenpairs of $(I_n - \tau \mathbf{1}\mathbf{1}^\top / n)A$. We now analyze what happens to these distinct sets of eigenvectors when applying the centering operation. Notice that the parameter $\tau$ controls how much of the mean is taken away and thus how much the centering influences the input graph. However, *as long as $\tau$ is not zero, there is always an effect altering the graph operator used in message-passing*:

**Proposition 5.1.** *Assuming $\tau > 0$,*

1. *$\mathcal{V}_{rest} \subset \hat{\mathcal{V}}$.*

2. *Assume that $A$ is not regular, then the dominant eigenvector $\nu$ of $A$ is **not** an eigenvector in $\hat{\mathcal{V}}$ for any eigenvalue.*

3. *$\sum_{(\lambda, \nu) \in \mathcal{V}_{struc}} \lambda > \sum_{(\hat{\lambda}, \hat{\nu}) \in \hat{\mathcal{V}} \backslash \mathcal{V}_{rest}} \hat{\lambda}$.*

The authors of GraphNorm Cai et al. (2021) had previously analysed the case for regular graphs. Here, we consider a general case where the graph is not regular, meaning there is more than one color in the final WL coloring $c^{(\infty)}$.

Assuming constant initialization of node features, $\mathcal{V}_{\text{struc}}$ spans the space of all possible node features that a GNN can compute and Proposition 5.1 states that it is exactly the space that the centering acts on. Specifically, while leaving the eigenvectors and eigenvalues in $\mathcal{V}_{\text{rest}}$ untouched, centering changes the eigenvector basis of the space spanned by $\mathcal{V}_{\text{struc}}$ in two ways: some vectors, such as the dominant eigenvector, are affected by this transformation and thus no longer convey the same information. At the same time, the centering transformation may change the magnitude of eigenvalues — that is, the dominant eigenvector may not be dominant anymore. Meanwhile, the whole space is pushed downward in the spectrum, meaning that after the centering transformation, the signal components within the structural eigenvectors are dampened and thus become less pronounced in the node representations given by the GNN.

Notably, such an effect altering the graph signal not only applies to BatchNorm with $\tau = 1$, but also GraphNorm (Cai et al., 2021), which was proposed specifically for GNNs. In their paper, the authors address the problem that BatchNorm's centering operation completely nullifies the graph signal on regular graphs. Their remedy is to only subtract an $\tau$ portion of the mean instead. However, Proposition 5.1 shows that a similar underlying problem altering the graph signal would persist for general graphs even switching from BatchNorm to GraphNorm.

**Comparison with standard neural networks** The use of normalization techniques in GNNs was inspired by use of normalization methods in standard feed-forward neural networks (Ioffe & Szegedy, 2015). Here, we want to emphasize that the issue centering causes in GNNs as described above is not a problem for standard neural networks, as standard neural networks do not need to incorporate the graph information in the forward pass. As a result, in standard neural networks, normalization transforms the input in a way that it does not affect the classification performance: for each neural network, there exists a neural network that yields an equivalent classification after normalization. However, this is not the case for GNNs. In GNNs, the graph information is added in through message-passing, which can be heavily altered by normalization as shown above. Such normalization can lead to information loss, negatively impacting the model's performance.

## 5.2 OUR METHOD: GRAPHNORMV2

Based on our theoretical analysis, we propose a new normalization layer for GNNs which has a similar motivation as the original GraphNorm but improves the centering operation to not affect the graph information in message-passing. Specifically, instead of naive centering, which can be thought of as subtracting a projection to the all-ones space, we use a learned projection. Learning a completely general projection may have certain downsides, however. As graphs can have different sizes, we either would need to learn different projections for different graph sizes or use only part of the learned projection for smaller graphs. We thus opt for learning a centering that transforms the features by subtracting an $(\tau_j)_i$ portion of the $i$-th eigenvector from the $j$-th column. Our proposed graph normalization is thus:

$$\text{GraphNormv2}(X_{:,j}) = \gamma_j \frac{X_{:,j} - (V_{k+}\tau_j\tau_j^\top V_{k+}^\top)X_{:,j}}{\sigma_j} + \beta_j,$$

Figure 1: **Long Term behavior of GCN.** Mean progression (over 10 independent trials) of $\mu_v(X^{(t)})$ and $\text{Rank}(X^{(t)})$ over 256 iterations of message-passing in both linear and non-linear GCN, where $v$ corresponds to the dominant eigenvector for $D^{-1/2}A_{\text{adj}}D^{-1/2}$. In the linear case, $\mu_v(X^{(t)})$ remains constant for all methods except the vanilla GCNs, indicating that complete collapse to the dominant eigenspace does not happen. However, PairNorm does collapse in terms of rank, while the other methods maintain a rank greater than 2. All the phenomena are explained by our theory. In the non-linear case, the models behave similarly. Notably, centering seems to prevent rank collapse in the non-linear case as PairNorm no longer collapses in rank.

where $\tau_j \in \mathbb{R}^{k+1}$ is a learnable parameter, $\sigma_j = \|X_{:,j} - (V_{k+}\tau_j\tau_j^\top V_{k+}^\top)X_{:,j}\|_2$, and $\gamma_j, \beta_j$ are the learnable affine parameters. Instead of just using the top-$k$ eigenvectors $V_k$ of the message-passing operator, we use $V_k$ and one additional vector $r = \mathbf{1} - V_k V_k^\dagger \mathbf{1}$ such that $\mathbf{1}$ can be represented as a linear combination. This ensures backward compatibility, in that, GraphNormv2 can emulate GraphNorm and BatchNorm. We denote the set $V_{k+} := V_k \cup \{r\}$.

## 6 NUMERICAL EXPERIMENTS

In this section, we investigate the benefits that can be derived from the proposed graph normalization method GraphNormv2. We examine the long term behavior of linear and non-linear GNNs by conducting an ablation study on randomly initialized, untrained GNNs. We then go on to inspect the practical relevance of our proposed method. More details about the experiments are provided in Appendix C.

**Rank collapse in linear and non-linear GNNs** We investigate the effect of normalization in deep (linear) GNNs on the Cora dataset (Yang et al., 2016). We employ seven different architectures: an architecture using residual connections, BatchNorm (Ioffe & Szegedy, 2015), PowerEmbed (Huang et al., 2022), PairNorm (Zhao & Akoglu, 2020), GraphNorm (Cai et al., 2021), GraphNormv2, and finally no normalization as a baseline. Each of these methods is used in an untrained, randomly initialized GCN (Kipf & Welling, 2017) (without biases) with 256 layers.

We compare these models using the following measures of convergence: $\mu_v(X^{(t)})$ as defined in (3), where $v$ is the dominant eigenvector, the eigenvector space projection: $d_{\text{ev}}(X) := \frac{1}{n}\|X - VV^\top X\|_F$, where $V \in \mathbb{R}^{n \times n}$ is the set of normalized eigenvectors of $A$, and the numerical rank of the features $\text{Rank}(X^{(t)})$. The results are shown in Figure 1. The same experiment with both GIN and GAT can be found in the appendix together with additional measures of convergence.

The two left panels of Figure 1 show that the commonly considered metric for measuring oversmoothing $\mu_v(X)$ indeed detects the collapse of node features in vanilla GNNs to the dominant eigenspace spanned by $v$ of the message-passing operator, in both the linear and non-linear case. Nonetheless, for any other methods, it does not detect any collapse of the feature space — neither in the linear nor non-linear case. However, the right two panels show PairNorm indeed also oversmoothes in the linear case. Specifically, it converges to the dominant eigenvector of $(I_n - \tau \mathbf{1}\mathbf{1}^\top/n)D^{-\frac{1}{2}}A_{\text{adj}}D^{-\frac{1}{2}})$. The other methods are able to preserve a rank greater than two over all iterations. However, they do converge to a low-dimensional subspace, e.g. PowerEmbed and GraphNormv2 converges to the top-$k$ eigenvector space as can be seen in the middle panel. All of these phenomena are explained by our theoretical analysis. As for the non-linear case, the models behave similarly apart from two things: (a) there is no convergence to the linear subspace due to the non-linearity $\sigma(\cdot)$, although the

Table 1: **Performance under different normalization layers.** Performance of GIN, GCN, and GAT with different normalization layers on graph classification task (MUTAG, PROTEINS, and PTC-MR) and node classification task (Cora, CiteSeer, and ogbn-arxiv). Results are reported as the mean accuracy (in %) ± std. over 10 independent trials and 5 folds. Best results are highlighted in blue; second best results are highlighted in gray.

| | | Graph Classification | | | Node Classification | | | Node Classification (# layers=20) | | |
|---|---|---|---|---|---|---|---|---|---|---|
| | | MUTAG | PROTEINS | PTC-MR | Cora | CiteSeer | ogbn-arxiv | Cora | CiteSeer | ogbn-arxiv |
| GCN | no norm | 78.2 ± 7.8 | 70.5 ± 4.0 | 57.7 ± 3.0 | 82.6 ± 4.6 | 89.4 ± 0.6 | 67.4 ± 0.2 | 46.0 ± 11.5 | 65.6 ± 4.3 | 6.0 ± 6.5 |
| | batch | 81.5 ± 2.0 | 70.4 ± 1.8 | 56.0 ± 3.5 | 85.2 ± 0.6 | 89.4 ± 0.7 | 68.3 ± 0.3 | 84.0 ± 1.6 | 82.2 ± 4.4 | 56.7 ± 0.5 |
| | graph | 81.1 ± 4.7 | 71.4 ± 3.6 | 58.1 ± 4.8 | 85.1 ± 0.8 | 89.1 ± 0.7 | 68.3 ± 0.2 | 80.5 ± 6.3 | 82.7 ± 2.1 | 56.3 ± 0.8 |
| | pair | 61.3 ± 10.1 | 59.6 ± 0.0 | 55.8 ± 0.0 | 84.2 ± 0.7 | 88.2 ± 0.7 | 63.0 ± 1.2 | 82.3 ± 0.5 | 74.0 ± 14.1 | 44.0 ± 1.9 |
| | group | 79.9 ± 5.8 | 69.3 ± 4.1 | 55.7 ± 4.1 | 85.5 ± 0.9 | 89.4 ± 0.6 | 65.1 ± 0.4 | 64.4 ± 10.9 | 80.7 ± 1.1 | 6.0 ± 6.2 |
| | powerembed | 82.4 ± 3.7 | 70.1 ± 3.6 | 55.2 ± 4.8 | 85.4 ± 0.8 | 89.7 ± 0.7 | 68.1 ± 0.2 | 48.0 ± 9.1 | 77.3 ± 3.8 | 55.9 ± 0.5 |
| | *graphv2* | 82.6 ± 4.6 | 72.6 ± 2.6 | 56.5 ± 4.8 | 85.8 ± 0.5 | 89.5 ± 0.6 | 68.3 ± 0.3 | 84.8 ± 0.8 | 83.4 ± 2.0 | 57.4 ± 0.6 |
| GAT | no norm | 78.5 ± 5.3 | 71.2 ± 2.0 | 61.3 ± 4.2 | 85.9 ± 0.9 | 93.7 ± 0.5 | 68.3 ± 0.9 | 79.2 ± 0.7 | 89.1 ± 0.5 | 44.6 ± 2.8 |
| | batch | 82.5 ± 4.1 | 68.4 ± 5.2 | 55.7 ± 0.7 | 86.2 ± 0.7 | 94.9 ± 0.5 | 71.0 ± 0.1 | 85.8 ± 0.7 | 94.2 ± 0.6 | 55.7 ± 0.5 |
| | graph | 81.2 ± 4.8 | 71.9 ± 2.4 | 60.3 ± 5.4 | 86.3 ± 0.9 | 94.8 ± 0.5 | 71.0 ± 0.1 | 85.7 ± 0.8 | 94.0 ± 0.6 | 55.9 ± 0.6 |
| | pair | 59.7 ± 11.1 | 60.1 ± 0.9 | 55.7 ± 1.1 | 85.4 ± 1.0 | 93.4 ± 1.2 | 70.0 ± 0.2 | 84.9 ± 0.7 | 92.5 ± 1.0 | 6.2 ± 0.9 |
| | group | 73.9 ± 3.6 | 69.1 ± 4.0 | 59.8 ± 3.8 | 87.4 ± 0.8 | 95.3 ± 0.5 | 70.9 ± 0.1 | 85.1 ± 0.7 | 94.2 ± 0.5 | 55.6 ± 0.6 |
| | powerembed | 75.0 ± 5.4 | 69.6 ± 4.0 | 58.4 ± 3.7 | 87.0 ± 0.8 | 95.5 ± 0.5 | 71.0 ± 0.1 | 85.4 ± 0.6 | 93.8 ± 0.4 | 55.6 ± 0.5 |
| | *graphv2* | 81.6 ± 3.8 | 71.0 ± 2.6 | 59.5 ± 4.4 | 86.3 ± 0.8 | 94.8 ± 0.5 | 70.9 ± 0.2 | 85.7 ± 0.8 | 94.4 ± 0.5 | 57.5 ± 0.6 |
| GIN | no norm | 79.7 ± 5.9 | 70.7 ± 3.6 | 59.2 ± 3.9 | 33.4 ± 45.4 | 47.0 ± 3.1 | 21.6 ± 0.0 | 28.2 ± 6.2 | 23.4 ± 5.1 | 25.7 ± 10.8 |
| | batch | 82.5 ± 5.0 | 69.2 ± 5.3 | 52.9 ± 5.3 | 68.4 ± 4.0 | 49.7 ± 6.8 | 21.1 ± 7.9 | 31.4 ± 5.4 | 23.6 ± 6.1 | 6.0 ± 6.1 |
| | graph | 83.7 ± 4.2 | 72.6 ± 2.4 | 59.1 ± 5.1 | 33.2 ± 3.4 | 63.0 ± 11.0 | 21.6 ± 0.0 | 26.9 ± 5.2 | 27.1 ± 4.6 | 6.2 ± 0.8 |
| | pair | 65.2 ± 3.2 | 64.5 ± 4.3 | 55.5 ± 1.6 | 37.2 ± 3.5 | 51.1 ± 9.2 | 20.7 ± 9.6 | 28.6 ± 6.2 | 26.9 ± 4.7 | 6.2 ± 0.9 |
| | group | 79.9 ± 5.8 | 69.3 ± 4.1 | 58.2 ± 2.7 | 85.9 ± 0.7 | 89.4 ± 0.6 | 20.1 ± 8.0 | 76.2 ± 9.1 | 94.0 ± 0.6 | 6.0 ± 8.1 |
| | powerembed | 82.4 ± 3.7 | 70.1 ± 3.6 | 59.8 ± 3.5 | 86.0 ± 1.1 | 89.7 ± 0.7 | 21.1 ± 6.5 | 40.0 ± 6.1 | 53.0 ± 3.6 | 6.0 ± 6.1 |
| | *graphv2* | 84.9 ± 3.6 | 71.0 ± 3.6 | 60.1 ± 5.5 | 86.5 ± 0.9 | 94.9 ± 0.5 | 68.9 ± 0.3 | 83.8 ± 2.0 | 93.8 ± 0.7 | 56.6 ± 0.7 |

rank can still be preserved. (b) The centering operation in PairNorm seems to prevent rank collapse in the non-linear case as PairNorm no longer collapses in rank.

**Classification performance** We evaluate the effectiveness of our method GraphNormv2 for real graph learning tasks. We perform graph classification tasks on the standard benchmark datasets MUTAG (Schlichtkrull et al., 2017), PROTEINS (Morris et al., 2020) and PTC-MR (Bai et al., 2019) as well as node classification tasks on Cora, Citeseer (Yang et al., 2016) and large-scale ogbn-arxiv (Hu et al., 2020). We compare GraphNormv2 to 5 normalization baselines: BatchNorm, PairNorm, GraphNorm, PowerEmbed and GroupNorm (Zhou et al., 2020). Following the general set-up of (Errica et al., 2019), we investigate the performance of GIN, GCN and GAT in a 5-fold cross-validation setting. Details on hyperparameter tuning and other specifics can be found in Appendix C. The final test scores are obtained as the mean scores across the 5 folds and 10 independent trials with the selected hyperparameters. We then also repeat the same experiment on Cora and Citeseer where we fix the depth of the models to 20 layers. The results are reported in Table 1.

Table 1 shows improvements on most benchmarks for our proposed normalization technique GraphNormv2. Although our method yields weak performance improvements in certain cases, this trend is apparent across datasets and tasks and even seems to be independent of the architecture. It is however worth mentioning that our method seems to not perform as well with GAT. This may be the result of GAT having both asymmetric and time-varying message-passing operators due to the attention mechanism (Wu et al., 2023a) — both aspects are outside the scope of our current analysis. More experimental results on other GNN backbones and heterophilic datasets can be found in Appendix D.

## 7 DISCUSSION

In this paper, we have analyzed the effect of both residual connections and normalization layers in GNNs. We show that both methods provably alleviate oversmoothing through the incorporation of the initial features and the scaling operation, respectively. In addition, by identifying that the centering operation in a normalization layer alters the graph information in message-passing, we proposed GraphNormv2, a novel normalization layer which does not distort the graph and empirically verified its effectiveness.

Our experiments showed that the trends described by our theoretical analysis are visible even in the non-linear case. Future work may concern itself with closing the gap and explaining how centering together with non-linearity can prevent node representations from collapsing to a low-dimensional subspace.

## ACKNOWLEDGEMENTS

Michael Scholkemper and Michael Schaub acknowledge partial funding from the DFG RTG 2236 "UnRAVeL" – Uncertainty and Randomness in Algorithms, Verification and Logic." and the Ministry of Culture and Science of North Rhine-Westphalia (NRW Ruckkehrprogramm). Xinyi Wu and Ali Jadbabaie were supported by ONR N00014-23-1-2299, ARO MURI W911 NF-19-1-0217, and a grant from Liberty Mutual.

## REPRODUCIBILITY

For the theoretical statements made in the main text, a number of assumptions are made. These are detailed in Section 3 and Section 4. All other needed assumptions are part of the respective statements. The proofs of all of these statements can be found in Appendix B.

The code is made available here. The results in Figure 1, can be reproduced by running the `ablation_study.ipynb` notebook in the supplementary material. Aggregating the statistics over multiple runs strengthens the reproducibility of the results.

The results in Table 1 can be reproduced by running the bash scripts `graph_level_norm.sh` and `node_level_norm.sh` respectively on a machine with a GPU. This will schedule **all** experiments reported in Table 1.

Further details regarding the experiments can be found in Appendix C.

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

# A  FURTHER BACKGROUND MATERIALS

## A.1  AN ALGEBRAIC PERSPECTIVE ON THE WEISFEILER-LEMAN COLORING

We briefly restate part of what has already been stated in the main text. So that notation is clear again. The Weisfeiler-Leman (WL) algorithm iteratively assigns a color $c(v) \in \mathbb{N}$ to each node $v \in V$ starting from a constant initial coloring $c^{(0)}(v) = 1 \ \ \forall v \in V$. In each iteration, an update of the following form is computed:

$$c^{(t+1)}(v) = \text{hash}\left(c^{(t)}(v), \{\{c^{(t)}(x) \mid x \in N(v)\}\}\right) \tag{6}$$

where hash is an injective hash function, and $\{\{\cdot\}\}$ denotes a multiset in which elements can appear more than once. The algorithm returns the final colors $c^{(\infty)}$ when the partition $\{(c^{(t)})^{-1}(c^{(t)}(v)) | v \in V\}$ no longer changes for consecutive $t$. citetmorris2019weisfeiler and citetxu2018powerful showed that GNNs cannot compute different features for nodes that are in the same class in the final coloring $c^{(\infty)}$.

For this paper, the equivalent algebraic perspective of the WL algorithm will be more useful: Given $c^{(\infty)}$ with $\text{Im}(c^{(\infty)}) = \{c_1, .., c_k\}$, define $H \in \{0,1\}^{n \times k}$ such that $H_{v,i} = 1$ if and only if $c^{(\infty)}(v) = c_i$. It holds that

$$AH = H(H^\top H)^{-1} H^\top AH = HA^\pi, \tag{7}$$

where $A^\pi := (H^\top H)^{-1} H^\top AH \in \mathbb{R}^{k \times k}$ is the adjacency matrix of the *quotient graph*. Looking at this equation, there are two things of interest here.

Firstly, $AH = HA^\pi$. Considering this equation more closely, the left side $AH$ counts, for each node, the number of neighbors of color $c_i$. More formally,

$$(AH)_{v,i} = \sum_{x \in (c^{(\infty)})^{-1}(c_i)} [x \in N(v)] = \sum_{x \in N(v)} [c(x) = c_i]$$

where the Iverson bracket $[\cdot]$ returns 1 if the statement is satisfied and 0 if it is not. The right hand side of the equation $(HA^\pi)$ states that nodes in the same class have the same rows. It is not hard to verify that if $c^{(\infty)}(v) = c_i$

$$(HA^\pi)_{v,:} = (A^\pi)_{i,:}$$

Now, combining both sides of the equation, the statement $AH = HA^\pi$ reads, that the number of neighbors of any color that a node has is the same for all nodes of the same class meaning $\{\{c^{(t)}(x) \mid x \in N(v)\}\}$ is the same for all nodes of the class. It becomes clear that at this point we have a fixed point of the WL update (6) and conversely whenever we have such a fixed point, the equation $AH = HA^\pi$ holds. The difference between the WL algorithm and (7) is that the latter equation holds for any so called *equitable partition* while the WL algorithm converges to the *coarsest* partition - meaning the partition with fewest distinct colors. For instance, on regular graphs, the WL

algorithm returns the partition with 1 color i.e. $H = \mathbb{1}$. However, the trivial partition with $n$ colors $H = I$ also fulfills (7).

Secondly $A^\pi := (H^\top H)^{-1}H^\top AH \in \mathbb{R}^{k \times k}$ is the adjacency matrix of the so-called *quotient graph*. Noticing that $H^\top H = \text{diag}(\{|(c^{(\infty)})^{-1}(c_i)|\}_{i \in [k]})$ it quickly becomes clear that $A^\pi$ is the graph of mean connectivity between colors. In other words, a supernode in the quotient graph represent a class of nodes in the original graph who share the same number of neighbors in each final coloring and an edge connecting two such supernodes is weighted by the number of edges there were going from nodes of the one color to nodes of the other color. In this sense, the quotient graph is a compression of the graph structure. It now only has $k$ supernodes compared to the $n$ nodes that there were in the original graph. Still the quotient graph is an adequate depiction of the structure of the graph. Most relevantly, the adjacency matrix $A$ of the original graph inherits all eigenpairs from the quotient graph: Let $(\lambda, \nu^\pi)$ be an eigenpair of $A^\pi$, then

$$AH\nu^\pi = HA^\pi\nu^\pi = \lambda H\nu^\pi .$$

We call such eigenvectors of $A$ the *structural eigenvectors*. They are profoundly important and may even completely determine processes that move over the edges of a network. The structural eigenvectors span a linear subspace that is invariant to multiplication with $A$. This means that once inside this subspace a graph dynamical system cannot leave it. This holds true for GNNs as Morris et al. (2019) and Xu et al. (2019) showed.

## A.2 UNIFYING PERSPECTIVES ON OVERSMOOTHING

Oversmoothing is a phenomenon observed in Graph Neural Networks (GNNs) where, as the number of layers increases, the node representations tend to become indistinguishable from each other. This phenomenon has its root cause in the underlying linear system, where it is known that, given mild assumptions that the system is *well-behaved* (ergodic and non-diverging), the linear system converges to the dominant eigenvector of the graph operator $A$.

In different analyses of the topic, different measures of convergence are used. Oono & Suzuki (2020) used the distance from the dominant eigenvector space

$$d_{\mathcal{M}}(X) = \inf\{\|X - Y\| \,|\, Y \in \mathcal{M}\} \tag{8}$$

where $\mathcal{M}$ is the dominant eigevector space. Cai & Wang (2020) used the graph Dirichlet energy

$$\mathcal{E}(X) = \frac{1}{2}\sum_{e \in E}\left(\frac{X_{i,:}}{\sqrt{d_i}} - \frac{X_{j,:}}{\sqrt{d_j}}\right)^2$$

Finally, Rusch et al. (2023); Wu et al. (2023a) used the distance to the all-ones vector

$$\mu(X) = \left\|X - \frac{\mathbf{1}\mathbf{1}^T}{n}X\right\|_2^2$$

While all of these measures of oversmoothing look different, the idea behind them is similar: They characterize the convergence of the features to a rank 1 matrix by showing that the respective functions $\varphi(X^{(t)}) \to \infty$ as $t \to \infty$, where $\varphi \in \{d_{\mathcal{M}}, \mathcal{E}, \mu(X)\}$. On can further strengthen the statement by showing that $\varphi(X^{(t)}) \leq C_1 e^{-C_2 t}$, i.e. that the features converge exponentially fast.

In this paper, we turn our analysis to $\mu_v$ as defined in Equation (3). We now show that each of the above definitions is equivalent to our in the sense that for any graph $G$ there exists a $v \in \mathbb{R}^n$ and $C_3, C_4 > 0$ such that

$$C_3\mu_v(X) \leq \varphi(X) \leq C_4\mu_v(X)$$

This is useful to our analysis, since $\varphi(X) = 0 \iff \mu_v(X) = 0$. Additionally, if we find a lower bound $\mu_v(X^{(t)}) > c$ (like in Proposition 4.1 and 4.6) this translates to a lower bound $\varphi(X^{(t)}) > c'$.

For $\varphi = \mu$ the equivalence is easy to see. Choosing $v = \mathbf{1}/\sqrt{n}$,

$$\mu_v = \left\|X - vv^T X\right\|_2^2 = \left\|X - \mathbf{1}\mathbf{1}^T/\sqrt{n}^2 X\right\|_2^2 = \mu(X)$$

So $C_3 = C_4 = 1$.

For $\varphi = d_{\mathcal{M}}$ this equivalence can be established if the graph is ergodic (strongly connected and aperiodic). Then, the dominant eigenspace is non-degenerate meaning there is a unique dominant eigenvector $\nu$ (up to scaling). Then,

$$d_{\mathcal{M}} = \inf\{\|X - Y\| \,|Y \in \mathcal{M}\} = \inf\{\|X - Y\| \,|Y_i \in \{a\nu | a \in \mathbb{R}\}\} = \left\|X - \nu\nu^T X\right\| = \mu_\nu(X)$$

So again, $C_3 = C_4 = 1$.

Lastly, for $\varphi = \mathcal{E}$, given, that the graph is connected, it is quite easy to see that $\mathcal{E}(X) = 0 \iff X \in \text{Span}(D^{\frac{1}{2}}\mathbf{1})$. Thus, we choose $v = D^{\frac{1}{2}}\mathbf{1}/\left\|D^{\frac{1}{2}}\mathbf{1}\right\|$. For simplicity, we show this for $X \in \mathbb{R}^n$ with $\|X\|_2 = 1$. The generalisation is straight forward. Let $d = \max_i D_{i,i}$:

$$\frac{\mu_v(X)}{2n^2\sqrt{d}} \leq \max_{(u,x)\in E}\left(\frac{|X_u - v_u| + |X_x - v_x|}{2\sqrt{d}}\right)^2$$

$$\leq \max_{(u,x)\in E}\left(|\frac{X_u}{2\sqrt{D_{u,u}}} - \frac{X_x}{2\sqrt{D_{x,x}}}|\right)^2$$

$$\leq \frac{1}{2}\sum_{e\in E}\left(\frac{X_{i,:}}{\sqrt{d_i}} - \frac{X_{j,:}}{\sqrt{d_j}}\right)^2 = \mathcal{E}(X) \leq \mu_v(X)$$

This shows that our findings (4.1, 4.6) that show that oversmoothing does not happen when using BatchNorm or residual connections carry over to GCN and attention-based GNNs for which the oversmoothing was shown to be a problem previously.

# B PROOFS

## B.1 PROOF FOR PROPOSITION 4.1

**Proposition 4.1.** *Let $v \in \mathbb{R}$ s.t. $\|v\|_2 = 1$. If $\mu_v(X^{(0)}) > 0$, then w.h.p $\exists c > 0$ s.t. $\mu_v(X^{(t)}) \geq c$.*

To prove this proposition, we first need to prove the following lemma:

**Lemma B.1.** *Let $q \in \mathbb{R}^n$ and $\|q^T X^{(0)}\|_2^2 > 0$ and let each $(W_l^{(t)})_{y,z} \overset{\text{i.i.d.}}{\sim} \mathcal{N}(\eta, s^2)$. Then for any $\epsilon > 0$, $\left\|q^\top X^{(t)}\right\|_2 \geq \alpha \cdot \epsilon \cdot \|q^T X^{(0)}\|_2^2$ with probability at least $p = 1 - \exp\left(-\frac{\epsilon^2}{2s^2}\right)$.*

*Proof.* We start by deconstructing $X^{(t)}$ as

$$X^{(t)} = (1 - \alpha)AX^{(t-1)}W_1^{(t-1)} + \alpha X^{(0)}W_2^{(t-1)}$$

Define $\hat{q} = \frac{1}{\|q^T X^{(0)}\|_2^2}q$. We then have:

$$\hat{q}^\top X^{(t)} = (1 - \alpha)\hat{q}^\top AX^{(t-1)}W_1^{(t-1)} + \alpha\hat{q}^\top X^{(0)}W_2^{(t-1)}$$
$$= \varphi + \alpha\hat{q}^\top X^{(0)}W_2^{(t-1)} \tag{9}$$

Resulting in:

$$\left\|\hat{q}^\top X^{(t)}\right\|_2 \geq \left\|\hat{q}^\top X^{(t)}\right\|_\infty = \left\|\varphi + \alpha\hat{q}^\top X^{(0)}W_2^{(t-1)}\right\|_\infty$$
$$= \max_j |\varphi_j + \alpha(\hat{q}^\top X^{(0)}W_2^{(t-1)})_j|$$
$$\geq |\varphi_j + \alpha(\hat{q}^\top X^{(0)}W_2^{(t-1)})_j|$$
$$= |\varphi_j + \alpha\hat{q}^\top (X^{(0)}W_2^{(t-1)})_{:,j}|$$
$$= |\varphi_j + \alpha\hat{q}^\top X^{(0)}(W_2^{(t-1)})_{:,j}| \tag{10}$$
$$= |\varphi_j + \alpha\sum_a(\hat{q}^\top)_a(X^{(0)})_{a,:}(W_2^{(t-1)})_{:,j}|$$
$$= |\varphi_j + \alpha\sum_b\sum_a(\hat{q}^\top)_a(X^{(0)})_{a,b}(W_2^{(t-1)})_{b,j}| = |\hat{Z}|$$

$\hat{Z}$ is a weighted sum of Gaussian variables and as such, is Gaussian itself ($\mathcal{N}(\hat{\eta}, \hat{s}^2)$) with mean $\hat{\eta} = \eta(\varphi_j) + \eta \sum_b \sum_a (\hat{q}^\top)_a (X^{(0)})_{a,b}$ and variance $\hat{s}^2 = s^2(\varphi_j) + \alpha^2 s^2 \sum_b (\sum_a (\hat{q}^\top)_a (X^{(0)})_{a,b})^2$. Because we defined $\hat{q} = \frac{1}{\|q^T X^{(0)}\|_2^2} q$, we have that $\sum_b (\sum_a (\hat{q}^\top)_a (X^{(0)})_{a,b})^2 = \|q^T X^{(0)}\|_2^2 \geq 1$ and as such $\hat{s}^2 \geq \alpha^2 s^2$.

Define the helper variable $\hat{Z}/\alpha$, which is Gaussian with $\hat{Z}/\alpha \sim \mathcal{N}(\frac{\hat{\eta}}{\alpha}, \frac{\hat{s}^2}{\alpha^2})$ and define the variable $Z \sim \mathcal{N}(0, s^2)$. Notice that $\hat{Z}/\alpha$ has higher variance than $Z$. To finish the proof, notice that

$$Pr(|\hat{Z}| \geq \alpha\epsilon) = Pr(|\hat{Z}|/\alpha \geq \epsilon) \geq Pr(|Z| \geq \epsilon) = 1 - Pr(|Z| < \epsilon) \geq 1 - \exp\left(-\frac{\epsilon^2}{2s^2}\right),$$

where the last inequality is based on the Chernoff Bound. To finish the proof, notice that:

$$\left\| \hat{q}^\top X^{(t)} \right\|_2 = \frac{\|q^\top X^{(t)}\|_2}{\|q^T X^{(0)}\|_2^2}$$

and thus:

$$Pr(\left\| \hat{q}^\top X^{(t)} \right\|_2 \geq \epsilon\alpha) \geq Pr(\frac{\|q^\top X^{(t)}\|_2}{\|q^T X^{(0)}\|_2^2} \geq \epsilon\alpha)$$

$$= Pr(\left\| q^\top X^{(t)} \right\|_2 \geq \epsilon\alpha \cdot \|q^T X^{(0)}\|_2^2) \geq 1 - \exp\left(-\frac{\epsilon^2}{2s^2}\right)$$

$\square$

We can now resume the proof of Proposition 4.1.

*Proof.* Let the weights be randomly initialized $(W_l^{(t)})_{y,z} \overset{\text{i.i.d.}}{\sim} \mathcal{N}(\eta, s^2)$. Assume

$$\mu_v(X^{(0)}) = \|(I - vv^\top)X^{(0)}\|_F^2 = \sum_{i=1}^k \|(I - vv^\top)X_{:,i}^{(0)}\|_2^2 > 0$$

There must be a column $X_{:,i}^{(0)}$ such that $X_{:,i}^{(0)}$ is not a scaled version of $v$. Equivalently, $X_{:,i}^{(0)} = av + br$, where $v^T r = 0$ and $(X_{:,i}^{(0)})^T r = b > 0$. Using Lemma B.1, we get that $\left\| r^\top X^{(t)} \right\|_2 \geq \alpha \cdot \epsilon \cdot \|r^T X^{(0)}\|_2^2$ with probability $p = 1 - \exp\left(-\frac{\epsilon^2}{2s^2}\right)$. By construction, $\|r^T X^{(0)}\|_2^2 > 0$ and we choose $\epsilon > 0$ such that $p = 0.9$. Thus, $\alpha \cdot \epsilon \cdot \|r^T X^{(0)}\|_2^2 = c > 0$. Lets now look at $\mu_v(X^{(t)})$. Since $\|v\| = 1$:

$$\|v^T X^{(t)}\|_F^2 \leq \left\| v^T X^{(t)} \right\|_2^2 - \left\| r^\top X^{(t)} \right\|_2^2 \leq \left\| v^T X^{(t)} \right\|_2^2 - c^2 \tag{11}$$

This means that

$$\left\| vv^T X^{(t)} \right\|_2^2 \leq \left\| X^{(t)} \right\|_2^2 - c^2$$

And thus:

$$\mu_v(X^{(t)}) = \left\| X^{(t)} - vv^T X^{(t)} \right\|_2^2 \geq \left\| X^{(t)} \right\|_2^2 - \left\| vv^T X^{(t)} \right\|_2^2 \geq c^2$$

$\square$

## B.2 PROPOSITION 4.1: DETERMINISTIC CASE

For the deterministic version, we adopt the following regularity conditions on weight matrices:

**Assumption B.2.** For the system described in (4), assume: there exists $\epsilon > 0$ such that

1. $\sum_{m=0}^t \alpha(1-\alpha)^m \lambda_i^m W_2^{(t-m)} W_1^{(t-m+1)} \cdots W_1^{(t)} + (1-\alpha)^{t+1} \lambda_i^{t+1} W_1^{(0)} \cdots W_1^{(t)}$ has smallest singular value $\sigma_{\min} \geq \epsilon$ for all $i \in [n]$.

2. $\sum_{m=0}^{t} \alpha(1-\alpha)^m \lambda_i^m W_2^{(t-m)} W_1^{(t-m+1)} \cdots W_1^{(t)} + (1-\alpha)^{t+1} \lambda_i^{t+1} W_1^{(0)} \cdots W_1^{(t)}$ converges as $t \to \infty$ and has smallest singular value $\sigma_{\min} \geq \epsilon$ for all $i \in [n]$.

Suppose $A$ is full-rank. If weights $W_1^{(t)}, W_2^{(t)}$ are orthogonal, then Assumption B.2.1 holds. On the other hand, Assumption B.2.2 is an asymptotic technical condition to ensure that weights are non-collapsing and non-diverging in the limit. Some ways to satisfy the assumptions is to have the spectral radius of $A$, $\rho(A) \leq 1$ and $W_1^{(t)}, W_2^{(t)} = I_k$ for any $t \geq 0$.

We restate Proposition 4.1 with full conditions: let $V \in \mathbb{R}^{n \times n}$ be the matrix of eigenvectors for $A$.

**Proposition B.3.** *Under Assumption B.2, let $\nu_q \in V$ be such that $\nu_q^\top v = 0$. If $X^{(0)}$ is properly initialized, such as if $X^{(0)}$ is not the zero matrix and $\|\nu_q^\top X^{(0)}\|_2 = c > 0$, then $\mu_v(X^{(t)}) \geq c\epsilon/\sqrt{k}$ for all $t \geq 0$ and $\lim_{t \to \infty} \mu_v(X^{(t)}) \geq c\epsilon/\sqrt{k}$.*

In particular, if $v \in V$ and and there exists $\nu_q \in V$ such that $\nu_q \neq v$ and $\|\nu_q^\top X^{(0)}\|_2 = c > 0$, meaning that if the initial feature $X^{(0)}$ contains a component in eigenvector $\nu_q$ other than $v$, that signal would always be maintained in $X^{(t)}$, even asymptotically.

*Proof.* Writing (4) recursively, we get that

$$X^{(t+1)} = \alpha \sum_{m=0}^{t} (1-\alpha)^m A^m X^{(0)} W_2^{(t-m)} W_1^{(t-m+1)} ... W_1^{(t)}$$
$$+ (1-\alpha)^{t+1} A^{t+1} X^{(0)} W_1^{(0)} ... W_1^{(t)} .$$

For each column $X_{:,i}^{(t+1)}$, similarly, one can prove by induction that

$$X_{:,i}^{(t+1)} = \sum_{l=1}^{n} \sum_{m=0}^{t} \sigma_{l,i}^{(t,m)} \lambda_l^m \nu_l ,$$

where

- $\Sigma^{(0,0)} = V^\top X^{(0)}$,

- $\Sigma^{(t,0)} = \alpha \Sigma^{(0,0)} W_2^{(t-1)}$ for all $t \geq 0$,

- $\Sigma^{(t,m)} = \alpha(1-\alpha)^m \Sigma^{(0,0)} W_2^{(t-m-1)} W_1^{(t-m)} ... W_1^{(t-1)}$, for all $1 \leq m \leq t-1$,

- $\Sigma^{(t,t)} = (1-\alpha)^t \Sigma^{(0,0)} W_1^{(0)} ... W_1^{(t-1)}$.

Then $\nu_q^\top X^{(t)}$

$$= \nu_q^\top X^{(0)} \left( \sum_{m=0}^{t} \alpha(1-\alpha)^m \lambda_i^m W_2^{(t-m)} W_1^{(t-m+1)} \cdots W_1^{(t)} + (1-\alpha)^{t+1} \lambda_i^{t+1} W_1^{(0)} \cdots W_1^{(t)} \right) .$$
$$(12)$$

Since by construction, $\|\nu_q^\top X^{(0)}\|_2 = c$, it follows from the regularity conditions on weights that

$$\|\nu_q^\top X^{(t)}\|_2 \geq c\epsilon .$$

This implies that

$$\|\nu_q^\top X^{(t)}\|_\infty \geq c\epsilon/\sqrt{k} ,$$

which means that there exists $i \in [k]$ such that

$$\left| \nu_q^\top X_{:,i}^{(t)} \right| = c\epsilon/\sqrt{k} .$$

Note that since $\nu_q \in V$ and $\nu_q^\top v = 0$, we get that

$$\mu_v(X^{(t)}) = \left\| X^{(t)} - vv^\top X^{(t)} \right\|_F = \sqrt{\sum_{l=1}^{k} \left\| X^{(t)}_{:,l} - vv^\top X^{(t)}_{:,l} \right\|_2^2}$$

$$\geq \sqrt{\left\| X^{(t)}_{:,i} - vv^\top X^{(t)}_{:,i} \right\|_2^2}$$

$$\geq \left| \nu_q^\top X^{(t)}_{:,i} \right|$$

which means that $\mu_v(X^{(t)}) \geq c\epsilon/\sqrt{k}$.

Similarly, we can show that $\lim_{t \to \infty} \mu_v(X^{(t)}) \geq c\epsilon/\sqrt{k}$. $\qquad \square$

### B.3 PROOF FOR PROPOSITION 4.2

**Proposition 4.2.** *Let* $x_i = X^{(0)}_{:,i}$, $\|x_i\|_2 = 1$ *and let each* $(W_l^{(t)})_{y,z} \overset{\text{i.i.d.}}{\sim} \mathcal{N}(\eta, s^2)$. *Then for any* $\epsilon > 0$, $\left\| x_i^\top X^{(t)} \right\|_2 \geq \epsilon$ *with probability at least* $p = 1 - \exp\left(-\frac{\epsilon^2}{2\alpha^2 s^2}\right)$.

*Proof.* Notice that $\left\| x_i^T X^{(0)} \right\|_2^2 \geq \left\| x_i^T X^{(0)}_{:,i} \right\|_2^2 = 1$. Using Lemma B.1, we get that $\left\| x_i^t X^{(t)} \right\|_2^2 > \alpha\epsilon$ with probability at least $p = 1 - \exp\left(-\frac{\epsilon^2}{2s^2}\right)$. From there it is easy to see that that $\left\| x_i^t X^{(t)} \right\|_2^2 > \epsilon$ with probability at least $p = 1 - \exp\left(-\frac{\epsilon^2}{2s^2\alpha^2}\right)$. $\qquad \square$

### B.4 PROPOSITION 4.2: DETERMINISTIC CASE

We complement 4.2 with the following result. Let $V \in \mathbb{R}^{n \times n}$ be the matrix of eigenvectors of $A$ and define

$$V^\star := \{\nu_q \in V : \left\| \nu_q^\top X^{(0)} \right\|_2 = c_q\}$$

$$V^0 := \{\nu_p \in V : \left\| \nu_p^\top X^{(0)} \right\|_2 = 0\},$$

where $c_q > 0$ for all $q$. In words, $V^\star$ is the set of eigenspaces of $A$ onto which $X^{(0)}$ has a non-trivial projection, and $V^0$ is the set of eigenspaces of $A$ onto which $X^{(0)}$ has no projection.

**Proposition B.4.** *Under Assumption B.2.1, for all $t \geq 1$,*

$$\left\| \nu_q^\top X^{(t)} \right\|_2 = c_q\epsilon, \forall \nu_q \in V^\star, \quad \left\| \nu_p^\top X^{(t)} \right\|_2 = 0, \forall \nu_p \in V^0.$$

*Under Assumption B.2.2,*

$$\lim_{t \to \infty} \left\| \nu_q^\top X^{(t)} \right\|_2 > c_q\epsilon, \forall \nu_q \in V^\star, \quad \lim_{t \to \infty} \left\| \nu_p^\top X^{(t)} \right\|_2 = 0, \forall \nu_p \in V^0.$$

*Proof.* The proof follows directly from the form (12). $\qquad \square$

The above result states that the signal excited in the original graph input $X^{(0)}$ is precisely what stays and the signal that is not excited in $X^{(0)}$ can never be created through message-passing.

We give the following concrete example of the above result:

**Example** Suppose $W_1^{(t)}, W_2^{(t)} = I$, then

$$X^{(t+1)} = \left( \alpha \sum_{k=0}^{t} (1-\alpha)^k A^k + (1-\alpha)^{t+1} A^{t+1} \right) X.$$

Note that when $\rho(A) < 1/(1-\alpha)$ such as $A = D^{-1/2}A_{\text{adj}}D^{-1/2}$, as $t \to \infty$,

$$\lim_{t \to \infty} X^{(t)} = \alpha(I_n - (1-\alpha)A)^{-1}X .$$

Let $(\lambda_i, \nu_i)$ be the $i$-th eigenpair of $A$ and $\sigma_{l,i} = \langle v_l, X_{:,i} \rangle$, then

$$X_{:,i}^{(t+1)} = \sum_{l=1}^{n} \left( \sum_{k=1}^{t} \alpha(1-\alpha)^k \lambda_l^k + (1-\alpha)^{t+1}\lambda_l^{t+1} \right) \sigma_{l,i}\nu_l ,$$

and when $\rho(A) < 1/(1-\alpha)$ such as $A = D^{-1/2}A_{\text{adj}}D^{-1/2}$,

$$\lim_{t \to \infty} X_{:,i}^{(t+1)} = \sum_{l=1}^{n} \frac{\alpha}{1 - (1-\alpha)\lambda_l} \sigma_{l,i}\nu_l .$$

This implies that for all $\nu_q$,

$$\|\nu_q^\top X^{(t)}\|_2 = \sqrt{\sum_{i=1}^{k} \left( \left( \sum_{m=1}^{t-1} \alpha(1-\alpha)^m \lambda_q^m + (1-\alpha)^t \lambda_q^t \right) \sigma_{q,i} \right)^2}$$

$$= \left( \sum_{m=1}^{t-1} \alpha(1-\alpha)^m \lambda_q^m + (1-\alpha)^t \lambda_q^t \right) \|\sigma_{q,:}\|_2$$

and as $t \to \infty$,

$$\lim_{t \to \infty} \|\nu_q^\top X^{(t)}\|_2 = \frac{\alpha}{1 - (1-\alpha)\lambda_q} \|\sigma_{q,:}\|_2 .$$

## B.5 PROOF OF PROPOSITION 4.3

**Proposition 4.3.** *Let* $\text{Kr}(A, X^{(0)}) = \text{Span}(\{A^{i-1}X_{:,j}^{(0)}\}_{i \in [n], j \in [k]})$ *be the Krylov subspace. Let* $Y \in \mathbb{R}^{n \times k}$. *Then there exist a* $T \in \mathbb{N}$ *and sequence of weights* $W_1^{(0)}, W_2^{(0)}, \cdots, W_1^{(T-1)}, W_2^{(T-1)}$ *such that* $X^{(T)} = Y$ *if and only if* $Y \in \text{Kr}(A, X^{(0)})$.

*Proof.* Set $T = n$, $\alpha = 0.5$ and $W_1^{(t)} = I$ for $t > 0$ and $W_1^{(0)} = \mathbb{0}$. Begin by unrolling the recursive equation (4):

$$X^{(1)} = (1-\alpha)AX^{(0)}W_1^{(0)} + \alpha X^{(0)}W_2^{(0)}$$

And in turn:

$$\begin{aligned} X^{(2)} &= (1-\alpha)AX^{(1)}W_1^{(1)} + \alpha X^{(0)}W_2^{(1)} \\ &= (1-\alpha)A((1-\alpha)AX^{(0)}W_1^{(0)} + \alpha X^{(0)}W_2^{(0)})I + \alpha X^{(0)}W_2^{(1)} \\ &= 0.5^2 AX^{(0)}W_2^{(0)} + 0.5X^{(0)}W_2^{(1)} \end{aligned}$$

Iterating this, yields:

$$X^{(n)} = \sum_{i=1}^{n} 0.5^i A^{i-1} X^{(0)} W_2^{(i-1)}$$

Now consider a single column of $X^{(n)}$:

$$\begin{aligned} (X^{(n)})_{:,j} &= \sum_{i=1}^{n} 0.5^i A^{i-1} X^{(0)} (W_2^{(i-1)})_{:,j} \\ &= \sum_{l=1}^{k} \sum_{i=1}^{n} 0.5^i (W_2^{(i-1)})_{l,j} A^{i-1} X_{:,l}^{(0)} \end{aligned}$$

Now let $Y_{:,j} \in \mathrm{Kr}(A, X^{(0)})$ be in the Krylov subspace. Then $Y_{:,j} = \sum_{l=1}^{k} \sum_{i=1}^{n} w_{l,i} A^{i-1} X_{:,l}^{(0)}$. Setting $(W_2^{(i-1)})_{l,j} = \frac{w_{l,i}}{0.5^i}$ yields $X_{:,j}^{(n)} = Y_{:,j}$.

For the other direction, begin similiarly by unrolling the recursive equation:

$$
\begin{aligned}
X^{(2)} &= (1-\alpha)AX^{(1)}W_1^{(1)} + \alpha X^{(0)}W_2^{(1)} \\
&= (1-\alpha)A((1-\alpha)AX^{(0)}W_1^{(0)} + \alpha X^{(0)}W_2^{(0)})W_1^{(1)} + \alpha X^{(0)}W_2^{(1)} \\
&= (1-\alpha)^2 A^2 X^{(0)}W_1^{(0)}W_1^{(1)} + (1-\alpha)\alpha AX^{(0)}W_2^{(0)}W_1^{(1)} + \alpha X^{(0)}W_2^{(1)}
\end{aligned}
$$

Iterating this, yields:

$$
X^{(n)} = \sum_{i=1}^{n+1} A^{i-1} X^{(0)} \mathcal{W}^{(i-1)}
$$

Now consider a single column of $X^{(n)}$:

$$
(X^{(n)})_{:,j} = \sum_{l=1}^{k} \sum_{i=1}^{n+1} A^{i-1} X_{:,l}^{(0)} \mathcal{W}_{l,j}^{(i-1)}
$$

Now, setting $w_{l,i} = \mathcal{W}_{l,j}^{(i-1)}$ and $Y_{:,j} = \sum_{l=1}^{k} \sum_{i=1}^{n+1} w_{l,i} A^{i-1} X_{:,l}^{(0)}$ verifies that $X_{:,j}^{(n)} = Y_{:,j} \in \mathrm{Kr}(A, X^{(0)})$. $\qquad \square$

### B.6 Proof for Proposition 4.6

**Proposition 4.6.** *Let $v \in \mathbb{R}^n$ s.t. $\|v\|_2 = 1$ and $v^T \mathbf{1} > 0$. Let $\mathrm{Rank}(V_{\neq 0}^\top X^{(0)}) > 1$, then there exists $c(v) > 0$ such that $\mu_v(X^{(t)}) \geq c(v)$ for all $t \geq 1$ with probability $1$.*

*Proof.* We prove this by induction. The base case for $0$ holds. Assume $\mathrm{Rank}(V_{\neq 0}^\top X^{(t)})$ has rank at least 2. Then, there exist at least 2 columns $X_{:,i}^{(t)}$ and $X_{:,j}^{(t)}$ such that $\mathrm{Rank}(V_{\neq 0}^\top \left[ X_{:,i}^{(t)}, X_{:,j}^{(t)} \right]) = 2$. Consider their eigenvector decomposition in terms of eigenvectors of $(I - \mathbf{1}\mathbf{1}^\top/n)A$:

$$
X_{:,i}^{(t)} = \sum_{l=1}^{n} \sigma_{l,i}^{(t)} v_l, \quad X_{:,j}^{(t)} = \sum_{l=1}^{n} \sigma_{l,j}^{(t)} v_l.
$$

Consider the action of $(I - \mathbf{1}\mathbf{1}^\top/n)A$:

$$
\tilde{X}_{:,i}^{(t)} = (I - \mathbf{1}\mathbf{1}^\top/n)AX_{:,i}^{(t)} = \sum_{l=1}^{n} \lambda_l \sigma_{l,i}^{(t)} v_l, \quad \tilde{X}_{:,j}^{(t)} = (I - \mathbf{1}\mathbf{1}^\top/n)AX_{:,j}^{(t)} = \sum_{l=1}^{n} \lambda_l \sigma_{l,j}^{(t)} v_l.
$$

Since $X_{:,i}^{(t)}$ and $X_{:,j}^{(t)}$ are linearly independent and the eigenvectors of a symmetric matrix are orthogonal, there exists $q$ such that $\sigma_{q,i}^{(t)} \neq \sigma_{q,j}^{(t)}$ with $\lambda_q \neq 0$. This exists because $\mathrm{Rank}(V_{\neq 0}^\top \left[ X_{i,:}^{(t)}, X_{j,:}^{(t)} \right]) = 2$. It follows that $\sigma_{q,i}^{(t)}\lambda_q \neq \sigma_{q,j}^{(t)}\lambda_q$, and thus the centered features $\tilde{X}_{:,i}^{(t)} \neq \tilde{X}_{:,j}^{(t)}$ and neither $\tilde{X}_{:,i}^{(t)} = \mathbf{0}$ nor $\tilde{X}_{:,j}^{(t)} = \mathbf{0}$ (otherwise $X_{:,i}^{(t)}$ would be a $0$ eigenvector and be orthogonal to $V_{\neq 0}$). Thus, they are linearly independent. Furthermore, $\tilde{X}_{:,i}^{(t)}, \tilde{X}_{:,j}^{(t)}, \mathbf{1}$ are linearly independent, since $(I - \mathbf{1}\mathbf{1}^\top/n)$ projects to the space orthogonal to $\mathrm{Span}\{\mathbf{1}\}$. Write:

$$
\begin{aligned}
X_{:,i}^{(t+1)} &= \frac{1}{\Gamma_i}(I - \mathbf{1}\mathbf{1}^\top/n)AX^{(t)}W_{:,i}^{(t)} \\
&= \frac{1}{\Gamma_i} \sum_{a=1}^{k} W_{a,i}^{(t)} \tilde{X}_{:,a}^{(t)} \\
&= \frac{1}{\Gamma_i} \sum_{a=1, a\neq i, a\neq j}^{k} W_{a,i}^{(t)} \tilde{X}_{:,a}^{(t)} + W_{i,i}^{(t)} \tilde{X}_{:,i}^{(t)} + W_{j,i}^{(t)} \tilde{X}_{:,j}^{(t)} \\
&= \frac{1}{\Gamma_i} (\varphi^{(i)} + W_{i,i}^{(t)} \tilde{X}_{:,i}^{(t)} + W_{j,i}^{(t)} \tilde{X}_{:,j}^{(t)}).
\end{aligned}
$$

We now consider the event that column $i$ collapses to the all-ones space. Notice that dividing the whole column by $\Gamma_i$ does not change whether or not the column has converged to the all-ones space or not. Thus,

$$
\begin{aligned}
\mathcal{A} &= \{W^{(t)} \in \mathbb{R}^{k \times k} \mid X^{(t+1)}_{:,i} = \beta 1\} \\
&= \{W^{(t)} \in \mathbb{R}^{k \times k} \mid \frac{1}{\Gamma_i}(\varphi^{(i)} + W^{(t)}_{i,i}\tilde{X}^{(t)}_{:,i} + W^{(t)}_{j,i}\tilde{X}^{(t)}_{:,j}) = \beta \mathbf{1}\} \\
&= \{W^{(t)} \in \mathbb{R}^{k \times k} \mid \varphi^{(i)} + W^{(t)}_{i,i}\tilde{X}^{(t)}_{:,i} + W^{(t)}_{j,i}\tilde{X}^{(t)}_{:,j} = \beta'\mathbf{1}\} \\
&= \{W^{(t)} \in \mathbb{R}^{k \times k} \mid W^{(t)}_{i,i}\tilde{X}^{(t)}_{:,i} + W^{(t)}_{j,i}\tilde{X}^{(t)}_{:,j} - \beta'\mathbf{1} = -\varphi^{(i)}\}
\end{aligned}
$$

Since $\tilde{X}^{(t)}_{:,i}, \tilde{X}^{(t)}_{:,j}, \mathbf{1}$ are linearly independent, given $\varphi^{(i)}$, there is only 1 solution to this equation. $\mathcal{A}$ is a proper hyperplane in $\mathbb{R}^{k \times k}$ and as such has Lebesgue measure 0. The event $\mathcal{A}$ thus has probability 0 and the opposite event, that column $i$ does not collapse to the all-ones space, has probability 1.

The same holds for $X^{(t+1)}_{:,j}$ and by the same reasoning, $X^{(t+1)}_{:,i}$ and $X^{(t+1)}_{:,j}$ are linearly independent with probability 1. Notice, that thus $\text{Rank}(V^\top_{\neq 0}\left[X^{(t+1)}_{:,i}, X^{(t+1)}_{:,j}\right]) = 2$ still holds. Notice that $\mathbf{1}^T X^{(t)} = 0$ because of the centering operation. However, since $\mathbf{1}^T v = b > 0$:

$$
\left\| v^T \left[X^{(t+1)}_{:,i}, X^{(t+1)}_{:,j}\right]\right\|^2_2 \leq \left\| \left[X^{(t+1)}_{:,i}, X^{(t+1)}_{:,j}\right]\right\|^2_2 - b^2
$$

This means that:

$$
\begin{aligned}
\mu_v(X^{(t+1)}) &\geq \mu_v(\left[X^{(t+1)}_{:,i}, X^{(t+1)}_{:,j}\right]) \\
&= \left\| \left[X^{(t+1)}_{:,i}, X^{(t+1)}_{:,j}\right] - vv^T \left[X^{(t+1)}_{:,i}, X^{(t+1)}_{:,j}\right]\right\| \\
&\geq \left\| \left[X^{(t+1)}_{:,i}, X^{(t+1)}_{:,j}\right]\right\| - \left\| vv^T \left[X^{(t+1)}_{:,i}, X^{(t+1)}_{:,j}\right]\right\| \\
&\geq b^2
\end{aligned}
$$

$\square$

## B.7 PROPOSITION 4.6: GENERAL CONDITIONS

We adopt the following regularity conditions:

**Assumption B.5.** For the system described in (5), assume:

1. For nonzero $x \in \mathbb{R}^n$ such that $x^\top \mathbf{1} = 0$, $Ax \notin \text{Span}\{\mathbf{1}\}$.

2. If $\mu_{\mathbf{1}}((AX^{(t)})_{:,i}) > 0$, then $\mu_{\mathbf{1}}((AX^{(t)}W^{(t)})_{:,i}) > 0$ for all $i \in [k]$.

Assumption B.5.1 ensure that the adversarial situation where one step of message-passing automatically leads to oversmoothing to the all-one subspace does not happen. Note that this is a more relaxed condition than requiring $\text{Span}\{\mathbf{1}\}^\perp$ to be an invariant subspace of $A$.

Assumption B.5.2 ensures that the case where weights are deliberately chosen for oversmoothing to happen in one layer does not happen, either. Note that for the second condition, such weights exist, i.e. let $W^{(t)} = I_k$. Moreover, if weights are randomly initialized, then Assumption B.5.2 holds almost surely.

We restate Proposition 4.6 under general conditions, which accounts for both linear and non-linear systems:

**Proposition B.6.** *For the system in (5), suppose $\sigma(\cdot)$ is injective and Assumption B.5 holds. Without loss of generality, also suppose the initial features $X^{(0)}$ are centered and all columns are nonzero. Then for $v \in \mathbb{R}^d$ where $v^\top \mathbf{1} \neq 0$, there exists $c(v) > 0$ such that $\mu_v(X^{(t)}) \geq c(v)\sqrt{k}$ for all $t \geq 0$.*

In particular, the most relevant case of our interest is that the message-passing $A$ corresponds to an graph operator such as the adjacency matrix (in the case of GIN), or the symmetric random walk

matrix $D^{-1/2}AD^{-1/2}$ (in the case of GCN) and $v$ is its dominant eigenvector. Under the assumption that the graph is connected, Perron-Frobenius Theorem implies $v$ is positive and hence $v^\top \mathbf{1} > 0$ holds.

We prove the statement below:

*Proof.* For each column in $X^{(0)}$, since it is centered and nonzero, $X^{(0)}_{:,i} \in \mathrm{Span}\{\mathbf{1}\}^\perp \backslash \{\mathbf{0}\}$ for all $i \in [k]$. Then given Assumption B.5.1, $AX^{(0)}_{:,i} \notin \mathrm{Span}\{\mathbf{1}\}$ and thus can be written as

$$AX^{(0)}_{:,i} = a\mathbf{1} + w_0^\perp \,,$$

where $a \in \mathbb{R}, w_0^\perp \in \mathrm{Span}\{\mathbf{1}\}^\perp$ and $w_0^\perp \neq \mathbf{0}$.

Then given Assumption B.5.2, $AX^{(0)}W^{(0)}_{:,i} \notin \mathrm{Span}\{\mathbf{1}\}$ and since $\sigma(\cdot)$ is injective, $\sigma(AX^{(0)}W^{(0)}_{:,i}) \notin \mathrm{Span}\{\mathbf{1}\}$ and

$$\sigma(AX^{(0)}W^{(0)}_{:,i}) = b\mathbf{1} + w_1^\perp \,,$$

where $b \in \mathbb{R}, w_1^\perp \in \mathrm{Span}\{\mathbf{1}\}^\perp$ and $w_1^\perp \neq \mathbf{0}$.

Then after the centering step of batch normalization,

$$(I - \mathbf{1}\mathbf{1}^\top/n)\sigma(AX^{(0)}W^{(0)}_{:,i}) = w_1^\perp \,,$$

and after the scaling step of batch normalization,

$$X^{(1)}_{:,i} = \mathrm{BN}(\sigma(AX^{(0)}W^{(0)}_{:,i})) = \frac{w_1^\perp}{\|w_1^\perp\|_2} \,,$$

which means each column of $X^{(1)} \in \mathrm{Span}\{\mathbf{1}\}^\perp \backslash \{\mathbf{0}\}$ and has norm 1. Note that the above argument applies for all $t \geq 0$ going from $X^{(t)}$ to $X^{(t+1)}$.

Let $v = c\mathbf{1}/\sqrt{n} + w_v^\perp$, where $c \neq 0, w_v^\perp \in \mathrm{Span}\{\mathbf{1}\}^\perp$ and $\|w_v^\perp\|_2^2 = 1 - c^2$. Notice that

$$
\begin{aligned}
\mu_v^2(X^{(t)}) &= \sum_{i=1}^k \|X^{(t)}_{:,i} - vv^\top X^{(t)}_{:,i}\|_2^2 = \sum_{i=1}^k \|X^{(t)}_{:,i}\|_2^2 - \|vv^\top X^{(t)}_{:,i}\|_2^2 \\
&= \sum_{i=1}^k \|X^{(t)}_{:,i}\|_2^2 - \|v(w_v^\perp)^\top X^{(t)}_{:,i}\|_2^2 \\
&\geq \sum_{i=1}^k 1 - \|w_v^\perp\|_2^2 = kc^2 \,,
\end{aligned}
$$

where the last inequality is because that $\|v(w_v^\perp)^\top X^{(t)}_{:,i}\|_2^2$ is maximized if $X^{(t)}_{:,i}$ and $w_v$ is aligned. We thus conclude that there exists $c(v) > 0$ such that $\mu_v(X^{(t)}) \geq c(v)\sqrt{k}$ for all $t \geq 0$. $\square$

## B.8 Proof for Proposition 4.8

**Proposition 4.8.** *Suppose $V_k^\top X^{(0)}$ has rank $k$, then for all weights $W^{(t)}$, the GNN with BatchNorm given in (5) exponentially converges to the column space of $V_k$.*

*Proof.* We will prove by induction on $t$ that

$$X^{(t)}_{:,i} = \frac{1}{\Gamma^{(t)}} \sum_{l=1}^n \sigma^{(t)}_{l,i} \lambda_l^t \nu_l \tag{13}$$

where $(\lambda_i, \nu_i)$ is the $i$-th eigenpair of $(I_n - \mathbf{1}\mathbf{1}^\top/n)A$ with $|\lambda_1| \geq |\lambda_2| \geq \cdots \geq |\lambda_n|$, $\Gamma^{(t)}$ is the normalization factor in the $t$-th round and $\sigma_{l,i}^{(t)} \in \mathbb{R}$. The base case follows from the decomposition of $X^{(0)}$ in the eigenvector basis of $(I_n - \mathbf{1}\mathbf{1}^T/n)A$: $X_{:,i}^{(0)} = \sum_{l=1}^n \langle X_{:,i}^{(0)}, \nu_l \rangle \nu_l$ (and the fact that $X^{(0)}$ is normalized).

For the induction step, the system can be rewritten as

$$X^{(t+1)} = (AX^{(t)}W^{(t)} - \mathbf{1}\mathbf{1}^\top AX^{(t)}W^{(t)}/n)\,\mathrm{diag}(..., \frac{1}{\mathrm{var}((AX^{(t)}W^{(t)})_{:,i})}, ...))$$

$$= (I_n - \mathbf{1}\mathbf{1}^\top/n)AX^{(t)}W^{(t)}D_{\mathrm{var}}\,.$$

Assuming (13),

$$((I_n - \mathbf{1}\mathbf{1}^\top/n)AX^{(t)})_{:,i} = \frac{1}{\Gamma^{(t)}}\sum_{l=1}^n \sigma_{l,i}^{(t)}\lambda_l^{t+1}\nu_l\,.$$

Further the action of $W^{(t)}$ is the following:

$$((I_n - \mathbf{1}\mathbf{1}^\top/n)AX^{(t)}W^{(t)})_{:,i} = \frac{1}{\Gamma^{(t)}}\sum_{j=1}^k W_{j,i}^{(t)}\sum_{l=1}^n \sigma_{l,j}^{(t)}\lambda_l^{t+1}\nu_l$$

$$= \frac{1}{\Gamma^{(t)}}\sum_{l=1}^n\sum_{j=1}^k (W_{j,i}^{(t)}\sigma_{l,j}^{(t)})\lambda_l^{t+1}\nu_l \tag{14}$$

$$= \frac{1}{\Gamma^{(t)}}\sum_{l=1}^n \sigma_{l,i}^{(t+1)}\lambda_l^{t+1}\nu_l\,.$$

Notice that $\sigma_{l,i}^{(t+1)} = \sum_{j=1}^k (W_{j,i}^{(t)}\sigma_{l,j}^{(t)})$ or equivalently, $\Sigma^{(t+1)} = \Sigma^{(t)}W^{(t)}$, where $\Sigma^{(t)} = [\sigma_{l,i}^{(t)}]$. Thus, $\Sigma^{(t+1)} = \Sigma^{(0)}W^{(0)}W^{(1)}\cdots W^{(t)}$. Lastly, the action of $D_{\mathrm{var}}$ is:

$$((I_n - \mathbf{1}\mathbf{1}^\top/n)AX^{(t)}W^{(t)}D_{\mathrm{var}})_{:,i} = \frac{1}{||\frac{1}{\Gamma^{(t)}}\sum_{l=1}^n \sigma_{l,i}^{(t+1)}\lambda_l^{t+1}\nu_l||_2}\frac{1}{\Gamma^{(t)}}\sum_{l=1}^n \sigma_{l,i}^{(t+1)}\lambda_l^{t+1}\nu_l$$

$$= \frac{1}{|\frac{1}{\Gamma^{(t)}}\sqrt{\sum_{l=1}^n(\sigma_{l,i}^{(t+1)}\lambda_l^{t+1})^2}|}\frac{1}{\Gamma^{(t)}}\sum_{l=1}^n \sigma_{l,i}^{(t+1)}\lambda_l^{t+1}\nu_l$$

$$= \frac{1}{|\sqrt{\sum_{l=1}^n(\sigma_{l,i}^{(t+1)}\lambda_l^{t+1})^2}|}\sum_{l=1}^n \sigma_{l,i}^{(t+1)}\lambda_l^{t+1}\nu_l$$

$$= \frac{1}{\Gamma^{(t+1)}}\sum_{l=1}^n \sigma_{l,i}^{(t+1)}\lambda_l^{t+1}\nu_l\,.$$

This concludes the induction. Notice that superscripts in brackets do not denote exponentiation, but rather an iterate at iteration $t$. Using this intermediate result (13), we show the following:

**Lemma B.7.** *For all $q > k$,*

$$||\nu_q^\top X^{(t)}||_2 \leq C_0 \left(\frac{\lambda_q}{\lambda_k}\right)^t\,.$$

Proving this directly yields Proposition 4.8. We have that $\Sigma^{(0)} = V^\top X^{(0)}$ due to the base case of the induction and by assumption, $\Sigma_{:k,:}^{(0)} = V_k^\top X^{(0)} \in \mathbb{R}^{k \times k}$ has rank $k$ and is therefore full rank. It is therefore invertible, meaning that there exists $(\Sigma_{:k,:}^{(0)})^{-1} \in \mathbb{R}^{k \times k}$ such that $V_k^\top X^{(0)}(\Sigma_{:k,:}^{(0)})^{-1} = I_k$.

We can thus write for the simplicity of notation:

$$\Sigma^{(0)} = \Sigma^{(0)}(\Sigma_{:k,:}^{(0)})^{-1}\Sigma_{:k,:}^{(0)} = \Sigma^{(\perp)}W^{(\perp)} = \begin{bmatrix} I_k \\ \Sigma_{(k+1):,:}^{(\perp)} \end{bmatrix} W^{(\perp)}\,.$$

This has the nice property that $\sigma_{i,i}^{(\perp)} = 1$ for $i \leq k$. Now, let's revisit (14) in that we can write $\Sigma^{(t+1)} = \Sigma^{(0)} W^{(0)} W^{(1)} \cdots W^{(t)}$. Let $\mathcal{W}^{(t)} = W^{(\perp)} W^{(0)} \cdots W^{(t)}$, meaning that $\sigma_{i,j}^{(t)} = \sum_{l=1}^{k} \mathcal{W}_{i,l}^{(t)} \sigma_{l,j}^{(\perp)}$. We now have everything to conclude the proof:

Consider now, the contribution of an eigenvector $\nu_q$ with $q > k$ at iteration $t$.

$$
\begin{aligned}
||\nu_q^\top X^{(t)}||_2 &= \sqrt{\sum_{i=1}^{k} (\nu_q^\top X_{:,i}^{(t)})^2} = \sqrt{\sum_{i=1}^{k} (\frac{1}{\Gamma_i^{(t)}} \sum_{l=1}^{n} \sigma_{l,i}^{(t)} \lambda_l^t \nu_q^\top \nu_l)^2} = \sqrt{\sum_{i=1}^{k} (\frac{1}{\Gamma_i^{(t)}} \sigma_{q,i}^{(t)} \lambda_q^t)^2} \\
&= \sqrt{\sum_{i=1}^{k} \frac{(\sigma_{q,i}^{(t)} \lambda_q^t)^2}{\sum_{p=0}^{n} (\sigma_{p,i}^{(t)} \lambda_p^t)^2}} = \sqrt{\sum_{i=1}^{k} \frac{(\sum_{l=1}^{k} \mathcal{W}_{l,i}^{(t)} \sigma_{q,l}^{(\perp)} \lambda_q^t)^2}{\sum_{p=1}^{n} (\sum_{l=1}^{k} \mathcal{W}_{l,i}^{(t)} \sigma_{p,l}^{(\perp)} \lambda_p^t)^2}} \\
&\leq \sqrt{\sum_{i=1}^{k} \frac{(\sum_{l=1}^{k} \mathcal{W}_{l,i}^{(t)} \sigma_{q,l}^{(\perp)} \lambda_q^t)^2}{\sum_{p=1}^{k} (\mathcal{W}_{p,i}^{(t)} \sigma_{p,p}^{(\perp)} \lambda_p^t)^2}} = \sqrt{\sum_{i=1}^{k} \frac{(\sum_{l=1}^{k} \mathcal{W}_{l,i}^{(t)} \sigma_{q,l}^{(\perp)} \lambda_q^t)^2}{\sum_{p=1}^{k} (\mathcal{W}_{p,i}^{(t)} \lambda_p^t)^2}} \\
&\leq \sqrt{\sum_{l=1}^{k} (\sigma_{q,l}^{(\perp)})^2 \sum_{i=1}^{k} \frac{\sum_{l=1}^{k} (\mathcal{W}_{l,i}^{(t)} \lambda_q^t)^2}{\sum_{p=1}^{k} (\mathcal{W}_{p,i}^{(t)} \lambda_p^t)^2}} \\
&\leq \sqrt{\sum_{l=1}^{k} (\sigma_{q,l}^{(\perp)})^2 \sum_{i=1}^{k} \frac{\sum_{l=1}^{k} (\mathcal{W}_{l,i}^{(t)} \lambda_q^t)^2}{\sum_{p=1}^{k} (\mathcal{W}_{p,i}^{(t)} \lambda_k^t)^2}} = \sqrt{\sum_{l=1}^{k} (\sigma_{q,l}^{(\perp)})^2 k \frac{\lambda_q^t}{\lambda_k^t}} \\
&\leq C_0 \frac{\lambda_q^t}{\lambda_k^t} .
\end{aligned}
$$

As we have that $\lambda_q < \lambda_k$ by construction, $||\nu_q^\top X^{(t)}||_2 \to 0$ exponentially as $t \to \infty$. $\qquad\square$

### B.9 PROOF FOR PROPOSITION 4.9

**Proposition 4.9.** *Suppose $|\hat{\lambda}_k| > 0$ and $V_k^\top X^{(0)}$ has rank $k$. For any $\epsilon > 0$, there exists $T > 0$ and a sequence of weights $W^{(0)}, W^{(1)}, ..., W^{(T)}$ such that for all $t \geq T$ and $i \in [k]$,*

$$
\left\| \nu_i^\top X_{:,i}^{(t)} \right\|_2 \geq 1/\sqrt{1+\epsilon} ,
$$

*where $\nu_i$ denotes the $i$-th eigenvector of $(I_n - \mathbf{1}\mathbf{1}^\top/n)A$.*

*Proof.* The proof idea is simple: we use Gaussian elimination to cancel out all "top-k" eigenvectors but the one in that row and then use the power iteration until the smaller eigenvectors are "drowned out" below the $\epsilon$ error margin. Assuming the columns of $X_{:k,:}^{(0)}$ are linearly independent (which is the case if it has rank $k$), this leads to the desired output: choose

$$
W_{j,i}^{(0)} = \begin{cases} 1 & \text{if } j = i \\ -\frac{\sigma_{1,i}^{(0)}}{\sigma_{1,1}^{(0)}} & \text{if } j = 1 \text{ and } i \neq 1 \\ 0 & \text{else} \end{cases}
$$

Then from (14) for each column $X_{:,i}$, we get that

$$
X_{:,i}^{(1)} = \frac{1}{\Gamma_i^{(1)}} \sum_{l=1}^{n} \sum_{j=1}^{k} (W_{j,i}^{(0)} \sigma_{l,j}^{(0)}) \lambda_l^1 \nu_l = \frac{1}{\Gamma_i^{(1)}} \sum_{l=1}^{n} (\sigma_{l,i}^{(0)} - \frac{\sigma_{1,i}^{(0)}}{\sigma_{1,1}^{(0)}} \sigma_{l,1}^{(0)}) \lambda_l^1 \nu_l .
$$

Now for $l = 1$, the factor $\sigma_{l,i}^{(0)} - \frac{\sigma_{1,i}^{(0)}}{\sigma_{1,1}^{(0)}} \sigma_{l,1}^{(0)} = 0$ yields

$$
X_{:,i}^{(1)} = \frac{1}{\Gamma_i^{(1)}} \sum_{l=2}^{n} (\sigma_{l,i}^{(0)} - \frac{\sigma_{1,i}^{(0)}}{\sigma_{1,1}^{(0)}} \sigma_{l,1}^{(0)}) \lambda_l^1 .
$$

Iterating this idea and choosing

$$
W_{j,i}^{(k)} = \begin{cases} 1 & \text{if } j = i \\ -\dfrac{\sigma_{k+1,i}^{(k)}}{\sigma_{k+1,k+1}^{(k)}} & \text{if } j = k+1 \text{ and } i \neq k+1 \\ 0 & \text{else}, \end{cases}
$$

we arrive at

$$
X_{:,i}^{(k)} = \frac{1}{\Gamma_i^{(k)}} \left( \sigma_{i,i}^{(i-1)} \lambda_i^k \nu_i + \sum_{l=k+1}^{n} \sigma_{l,i}^{(k)} \lambda_l^k \nu_l \right).
$$

Now, we switch gears and use $W^{(t)} = I_n$ for $t \geq k$, it follows that

$$
||\nu_i^\top (X_{:,i}^{(t)})||_2 = \frac{\sigma_{i,i}^{(i-1)} \lambda_i^t}{\sqrt{(\sigma_{i,i}^{(i-1)} \lambda_i^t)^2 + \sum_{l=k+1}^{n} (\sigma_{l,i}^{(k)} \lambda_l^t)^2}}
$$

$$
= \frac{1}{\sqrt{1 + \frac{\sum_{l=k+1}^{n} (\sigma_{l,i}^{(k)} \lambda_l^t)^2}{(\sigma_{i,i}^{(i-1)} \lambda_i^t)^2}}}.
$$

The statement left to prove is thus:

$$
\epsilon \geq \frac{\sum_{l=k+1}^{n} (\sigma_{l,i}^{(k)} \lambda_l^t)^2}{(\sigma_{i,i}^{(i-1)} \lambda_i^t)^2}
$$

$$
\Longleftarrow \epsilon \geq \frac{(n-k) \max_l (\sigma_{l,i}^{(k)})^2}{(\sigma_{i,i}^{i-1})^2} \frac{\lambda_{k+1}^{2t}}{\lambda_i^{2t}}
$$

$$
\Longleftarrow \frac{\log\left( \frac{\epsilon \sigma_{i,i}^{(i-1)}}{(n-k) \max_l (\sigma_{l,j}^{(k)})^2} \right)}{2 \log\left( \frac{\lambda_{k+1}}{\lambda_i} \right)} \leq t.
$$

Thus, setting $T$ to be larger than this bound yields the desired claim. $\qquad \square$

## B.10  PROOF FOR PROPOSITION 5.1

**Proposition B.8.** *Let $A \in \mathbb{R}_{\geq 0}^{n \times n}$ be a symmetric non-negative matrix. Let $H \in \{0,1\}^{n \times m}$ such that $AH = HA^\pi$ is the coarsest EP of $A$. Divide the eigenpairs $\mathcal{V} = \{..., (\lambda, \nu), ...\}$ of $A$ into the following two sets: $\mathcal{V}_{struc} = \{(\lambda, \nu) \in \mathcal{V} \mid \nu = H\nu^\pi\}, \mathcal{V}_{rest} = \mathcal{V} \backslash \mathcal{V}_{struc}$. Let $\hat{\mathcal{V}} = \{..., (\hat{\lambda}, \hat{\nu}), ...\}$ be eigenpairs of $(I_n - \tau \mathbf{1}\mathbf{1}^\top/n)A$, for $\tau \neq 0$. Then*

1. *$\mathcal{V}_{rest} \subset \hat{\mathcal{V}}$.*

2. *Assume $\mathcal{V}_{struc} \neq \{(\lambda, \mathbf{1})\}$. Let $(\lambda, \nu)$ be the dominant eigenpair of $A$. Then $\nu$ is **not** an eigenvector of $(I_n - \tau \mathbf{1}\mathbf{1}^\top/n)A$.*

3. *$\sum_{(\lambda, \nu) \in \mathcal{V}_{struc}} \lambda > \sum_{(\hat{\lambda}, \hat{\nu}) \in \hat{\mathcal{V}} \backslash \mathcal{V}_{rest}} \hat{\lambda}$*

*Proof.* Let $H \in \{0,1\}^{n \times m}$ indicate the coarsest EP of $A$ ($AH = HA^\pi$). As each node belongs to exactly 1 class, it holds that $H\mathbf{1}_n = \mathbf{1}_m$. We prove that for any eigenpair $(\lambda, \nu) \in \mathcal{V}_{rest}$, it holds that $\nu^\top \mathbf{1} = 0$. From this, the first statment follows quickly:

$$
(I_n - \tau \mathbf{1}\mathbf{1}^T/n)A\nu = A\nu - \tau \mathbf{1}\mathbf{1}^T A\nu/n = \lambda\nu - \tau\lambda \mathbf{1}\mathbf{1}^T \nu/n = \lambda\nu.
$$

To prove this, let's look at $A^\pi$. $A^\pi$ is not symmetric, but its eigenpairs are associated to $A$'s eigenpairs in the following way. Let $(\lambda, \nu^\pi)$ be an eigenpair of $A^\pi$, then $(\lambda, H\nu^\pi)$ is an eigenpair of $A$: $AH\nu^\pi = HA^\pi = H\lambda\nu^\pi$. As $A$ is a symmetric matrix, its eigenvectors are orthogonal implying $(H\nu_i^\pi)^\top (H\nu_j^\pi) = 0$.

**Lemma B.9.** *The eigenvectors $\nu_1^\pi, ..., \nu_m^\pi$ of $A^\pi$ are linearly independent.*

For simplicity, choose the eigenvectors $\nu_i^\pi$ to be normalized in such a way, that $(H\nu_i^\pi)^\top(H\nu_i^\pi) = 1$. Suppose for a contradiction, that they are linearly dependent and without loss of generality, $\nu_m^\pi = \sum_{i=1}^{m-1} a_i \nu_i^\pi$. Take $j$ such that $a_j \neq 0$, this must exist otherwise $\nu_m = \mathbf{0}$. Now,

$$(H\nu_m^\pi)^\top(H\nu_j^\pi) = (H\sum_{i=1}^{m-1} a_i \nu_i^\pi)^\top (H\nu_j^\pi) = \sum_{i=1}^{m-1}(Ha_i\nu_i^\pi)^\top(H\nu_j^\pi) = a_j \neq 0\,,$$

which is a contradiction.

Since the eigenvectors of $A^\pi$ are linearly independent, there exists a unique $\beta \in \mathbb{R}^m$ s.t. $\sum_{i=1}^m \beta_i \nu_i^\pi = \mathbf{1}_m$. Finally let $\varphi \in \mathcal{V}_{\text{rest}}$,

$$\varphi^\top \mathbf{1}_n = \varphi^\top H \mathbf{1}_m = \varphi^\top H \sum_{i=1}^m \beta_i \nu_i^\pi = \sum_{i=1}^m \beta_i \varphi^\top H \nu_i^\pi = 0\,.$$

This concludes the proof of the first statement.

To prove the second statement, let $(\lambda, \nu)$ be a dominant eigenpair of $A$, such that $\nu \geq 0$ is non-negative. This exists as $A$ is a non-negative matrix. Additionally, $\nu$ is not the all-zeros vector and as such $\mathbf{1}^\top \nu > 0$. By assumption $\nu \neq \mathbf{1}$. Then:

$$(I - \tau \mathbf{1}\mathbf{1}^\top/n)A\nu = \hat{\lambda}\nu \iff \lambda\nu - \tau\mathbf{1}\mathbf{1}^\top\lambda\nu/n = \hat{\lambda}\nu \iff (\lambda - \hat{\lambda})\nu - \tau\lambda\mathbf{1}\mathbf{1}^\top\nu/n = 0\,.$$

As $\nu \neq \mathbf{1}$, there exist $i, j$ s.t. $\nu_i \neq \nu_j$ thus, $(\lambda - \hat{\lambda})\nu_i - \tau\lambda(\mathbf{1}\mathbf{1}^\top\nu)_i/n = 0$ and $(\lambda - \hat{\lambda})\nu_j - \tau\lambda(\mathbf{1}\mathbf{1}^\top\nu)_j/n = 0$ cannot be true at the same time. Thus, this equation has no solution and $\nu$ is not an eigenvector of $(I_n - \tau\mathbf{1}\mathbf{1}^\top/n)A$.

For the third and final statement, notice that the trace of $A$ is larger than the trace of $(I_n - \tau\mathbf{1}\mathbf{1}^\top/n)A$. Since the trace of a matrix is the sum of its eigenvalues, we have

$$\sum_{(\lambda,\nu)\in\mathcal{V}} \lambda = \text{Tr}(A) > \text{Tr}((I_n - \tau\mathbf{1}\mathbf{1}^\top/n)A) = \sum_{(\hat{\lambda},\hat{\nu})\in\hat{\mathcal{V}}} \hat{\lambda}\,.$$

Consequently,

$$\sum_{(\lambda,\nu)\in\mathcal{V}} \lambda = \sum_{(\lambda,\nu)\in\mathcal{V}_{\text{struc}}} \lambda + \sum_{(\lambda,\nu)\in\mathcal{V}_{\text{rest}}} \lambda > \sum_{(\hat{\lambda},\hat{\nu})\in\hat{\mathcal{V}}\backslash\mathcal{V}_{\text{rest}}} \hat{\lambda} + \sum_{(\lambda,\nu)\in\mathcal{V}_{\text{rest}}} \lambda = \sum_{(\hat{\lambda},\hat{\nu})\in\hat{\mathcal{V}}} \hat{\lambda}\,.$$

Subtracting $\sum_{(\lambda,\hat{\nu})\in\mathcal{V}_{\text{rest}}} \lambda$ from both sides yields the final statement.

$\square$

## C  Experiments

We compare these models using the measures of convergence: $\mu_v(X^{(t)})$ as defined in (3), where in this case for GATs and GINs, the dominant eigenvector $v$ is $\mathbf{1}/\sqrt{n}$, the numerical rank of the features $\mathrm{Rank}(X^{(t)})$, the column distance used in (Zhao & Akoglu, 2020):

$$d_{\mathrm{col}}(X) := \frac{1}{d^2} \sum_{i,j} \left\| \frac{X_{:,i}}{\|X_{:,i}\|_1} - \frac{X_{:,j}}{\|X_{:,j}\|_1} \right\|_2,$$

the column projection distance:

$$d_{\mathrm{p\text{-}col}}(X) := \frac{1}{d^2} \sum_{i,j} 1 - \frac{X_{:,i}^\top}{\|X_{:,i}\|_2} \frac{X_{:,j}}{\|X_{:,j}\|_2},$$

and the eigenvector space projection:

$$d_{\mathrm{ev}}(X) := \frac{1}{n} \left\| X - VV^\top X \right\|_F,$$

where $V \in \mathbb{R}^{n \times n}$ is the set of normalized eigenvectors of $A$, and finally, the rank of $X$: $\mathrm{Rank}(X)$.

**Datasets**  We provide summary statistics for datasets used in experiments in Section 6 in Table 2.

Table 2: The summary statistics of the datasets used in Section 6.

| Dataset | #graphs | #nodes | #edges | #features | # classes |
|---|---|---|---|---|---|
| **MUTAG** | 188 | ~17.9 | ~39.6 | 7 | 2 |
| **PROTEINS** | 1,113 | ~39.1 | ~145.6 | 3 | 2 |
| **PTC-MR** | 344 | ~14.29 | ~14.69 | 18 | 2 |
| **Cora** | 1 | 2,708 | 10,556 | 1,433 | 7 |
| **Citeseer** | 1 | 3,327 | 9,104 | 3,703 | 6 |
| **ogbn-arxiv** | 1 | 169,343 | 1,166,243 | 128 | 40 |

**Training details**  We perform a within-fold 90%/10% train/validation split for model selection. We train the models for 200 epochs using the AdamW optimizer and search the hyperparameter space over the following parameter combinations:

- learning rate $\in \{10^{-4}, 10^{-3}, 10^{-2}, 10^{-1}\}$
- feature size $\in \{32, 64\}$
- weight decay $\in \{0, 10^{-2}, 10^{-4}\}$
- number of layers $\in \{3, 5\}$

We select the hyperparameters of the model with the best mean validation accuracy over its 30 best epochs. The code and all non publicly available data is available here.

**Compute**  We ran all of our experiments on a system with two NVIDIA L40 GPUs, two AMD EPYC 7H12 CPUs and 1TB RAM.

**Licenses**

- PyG (Fey & Lenssen, 2019): MIT license
- OGB (Hu et al., 2020): MIT license
- ogbn-arxiv: ODC-BY

# D ADDITIONAL EXPERIMENTAL RESULTS

## D.1 LONG-TERM MODEL PERFORMANCE

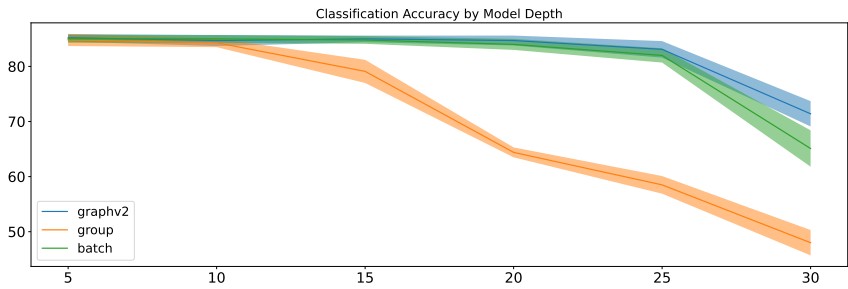

Figure 2: **Long-term behavior of GCN performance.** Classification accuracy and standard deviation of GCN models of varying depth. The x-axis show the depth of the GCN while the y axis shows classification accuracy and standard deviations.

## D.2 NODE CLASSIFICATION PERFORMANCE ON HETEROPHILIC GRAPHS

It is worth noting that our theoretical results in the paper hold for any initial features and any target labels and thus hold for datasets exhibiting either homophily or heterophily and even those that are not necessarily either of the two. In particular, the graph-level tasks we present in Section 6 can be seen as heterophilic (in terms of nodes features while the targets are neither). We can see in those cases, GraphNormv2 also performs well.

For the node classification task, we conduct additional node classification experiments using GCN as the backbone on the heterophilic dataset Cornell Pei et al. (2020), using the default splits provided in PyG. The results are shown in Table 3. We see that our method, GraphNormv2 still performs competitively with baseline methods and outperforms the baselines in the deep architecture regime.

Table 3: **Performance on heterophilic graphs.** Performance of GCN with different normalization layers on the heterogeneous node classification dataset Cornell. Results are reported as the mean accuracy (in %) $\pm$ std. over 10 independent trials and the 10 fixed splits. Best results are highlighted in blue; second best results are highlighted in gray.

|     |            | Node Classification | Node Classification (# layers=20) |
| --- | ---------- | ------------------- | --------------------------------- |
|     |            | Cornell             | Cornell                           |
|     | no norm    | $50.9 \pm 6.2$      | $47.4 \pm 4.4$                    |
|     | batch      | $50.1 \pm 4.4$      | $45.7 \pm 4.9$                    |
|     | graph      | $52.8 \pm 3.5$      | $48.6 \pm 4.6$                    |
| GCN | pair       | $47.0 \pm 5.2$      | $44.9 \pm 4.0$                    |
|     | group      | $52.8 \pm 4.0$      | $49.8 \pm 4.4$                    |
|     | powerembed | $50.8 \pm 5.4$      | $48.8 \pm 5.4$                    |
|     | *graphv2*  | $52.2 \pm 4.2$      | $51.0 \pm 4.2$                    |

Table 4: **Performance of GraphSAGE under different normalization layers.** Performance of GraphSAGE with different normalization layers on the node classification dataset Cora. Results are reported as the mean accuracy (in %) $\pm$ std. over 10 independent trials and 5 folds. Best results are highlighted in blue; second best results are highlighted in gray.

| | | Node Classification | Node Classification (# layers=20) |
|---|---|---|---|
| | | Cora | Cora |
| | no norm | $85.0 \pm 0.7$ | $83.3 \pm 1.2$ |
| | batch | $32.1 \pm 5.8$ | $30.1 \pm 3.8$ |
| | graph | $30.9 \pm 4.7$ | $30.1 \pm 3.8$ |
| GraphSAGE | pair | $32.7 \pm 4.3$ | $29.7 \pm 5.2$ |
| | group | $87.2 \pm 0.7$ | $85.6 \pm 0.9$ |
| | powerembed | $86.3 \pm 0.8$ | $85.2 \pm 0.6$ |
| | *graphv2* | $85.1 \pm 0.9$ | $85.9 \pm 0.8$ |

## D.3 NODE CLASSIFICATION PERFORMANCE WITH GRAPHSAGE BACKBONE

For the node classification task, we conduct additional node classification experiments using Graph-SAGE Hamilton et al. (2017) as the backbone under the same cross-validation setup specified in the paper (see Section 6). The results are shown in Table 4.

Our method, GraphNormv2 still performs competitively with baseline methods and outperforms the baselines in the deep architecture regime. Note that the message-passing operator for GraphSAGE is a row-stochastic matrix, hence its dominant eigenvector is the all-ones vector. We suspect that this might be the case why BatchNorm and GraphNorm perform so poorly with this backbone, as their centering step uses the all-ones vector and thus distorts the graph signal as discussed in Section 5 of the paper.

