# OpenReview forum: "Residual Connections and Normalization Can Provably Prevent Oversmoothing in GNNs"
_ICLR.cc/2025/Conference — ICLR 2025 Poster_

### Official Review · Reviewer_JBkc · 2024-10-31

**Soundness:** 3
**Presentation:** 3
**Contribution:** 3
**Rating:** 6
**Confidence:** 3

**Summary:**

This paper provides a formal and precise theoretical analysis of linearized Graph Neural Networks (GNNs) with residual connections and normalization layers. Notably, it reveals how the centering step in normalization can modify the message-passing mechanism of graph signals. Building on this insight, the authors propose a novel, theoretically-grounded normalization layer, GraphNormv2. Experimental results support the efficacy of GraphNormv2, showing its ability to enhance performance across diverse GNN architectures and tasks.

**Strengths:**

The paper presents valuable contributions by providing a theoretical foundation that enhances understanding of existing techniques for addressing the oversmoothing problem and informs the design of novel normalization layers in GNNs, notably GraphNormv2. The work is well-written, concise, logically structured, and easy to follow, making it accessible to a broad audience. Additionally, the experimental evidence supports the effectiveness of GraphNormv2 across various GNN architectures and tasks.

**Weaknesses:**

The experimental results are limited and do not sufficiently support the paper's claims, particularly regarding the effectiveness of GraphNormv2 across diverse GNN architectures and tasks.

1. The authors test their method solely on homophilic graphs, limiting the generalizability of their findings. Is there any rationale reason for choosing only homophilic datasets? The oversmoothing problem is particularly pronounced in heterophilic graphs, so it would be valuable to evaluate GraphNormv2 on heterophilic datasets as well. I suggest including experiments on heterophilic datasets such as Texas, Cornell, and at least one large-scale dataset, like Arxiv-year, to better demonstrate the robustness and effectiveness of GraphNormv2 across diverse graph structures.

2. Additionally, whether the theoretical analysis suggests any potential challenges or differences in applying GraphNormv2 to heterophilic graphs compared to homophilic ones?

3. The proposed method is tested on a limited range of GNN architectures, specifically GCN, GAT, and GIN. However, other types of GNNs, such as GraphSAGE and diffusion-based models like Graph Diffusion Networks (GDNs) and Diffusion Convolutional Neural Networks (DCNN), are also susceptible to oversmoothing. So, whether theoretical analysis does suggest any potential challenges or differences in applying GraphNormv2 to heterophilic graphs compared to homophilic ones? If not, testing GraphNormv2 on a broader set of GNN architectures would provide a more comprehensive evaluation and further validate its effectiveness across diverse model types.

**Questions:**

Please see the weakness.

---

> ### Author Response · Authors · 2024-11-21
> **Response to Reviewer JBkc**
>
> Thank you for your constructive feedback and positive assessment of our work. In line with the reviewer’s suggestion, we have conducted additional sets of experiments on the node classification task on heterophilic graphs and on the node classification task with GraphSAGE backbone. Below, we provide individual responses to your questions and comments.
>
> **Q1: The authors test their method solely on homophilic graphs, limiting the generalizability of their findings.**
>
> Thank you for the comment. In line with the reviewer’s suggestion, we have conducted additional node classification experiments using GCN as the backbone on the heterophilic dataset Cornell, using the default splits provided in PyG. We would like to remark that the amount of additional experiments that could be performed was limited due to time constraints for the rebuttal. For this reason, we could not perform the experiments on a larger scale dataset, which we will aim to complete for camera ready submission. The results are shown in the following table:
>
> |            | Node Classification | Node Classification (#layers=20) |
> |------------|---------------------|--------------------------------------|
> |            | Cornell             | Cornell                              |
> | no norm    | $50.9 \pm 6.2 $     | $47.4 \pm 4.4 $                      |
> | batch      | $50.1 \pm 4.4 $     | $45.7 \pm 4.9 $                      |
> | graph      | $\mathbf{52.8 \pm 3.5}$     | $48.6 \pm 4.6 $                      |
> | pair       | $47.0 \pm 5.2 $     | $44.9 \pm 4.0 $                      |
> | group      | $\mathbf{52.8 \pm 4.0}$     | $49.8 \pm 4.4 $                      |
> | powerembed | $50.8 \pm 5.4 $     | $48.8 \pm 5.4 $                      |
> | graph2 | $52.2 \pm 4.2 $     | $\mathbf{51.0 \pm 4.2}$                       |
>
> Our method, GraphNormv2 still performs competitively with baseline methods and outperforms the baselines in the deep architecture regime.
>
> **Q2: Additionally, whether the theoretical analysis suggests any potential challenges or differences in applying GraphNormv2 to heterophilic graphs compared to homophilic ones?**
>
> Thank you for the question. Our theoretical results hold for any initial features and any target labels and thus hold for datasets exhibiting either homophily or heterophily and even those that are not necessarily either of the two. In particular, the graph-level tasks we present can be seen as heterophilic (in terms of nodes features while the targets are neither). We can see in those cases, GraphNormv2 also performs well.
>
> **Q3: The proposed method is tested on a limited range of GNN architectures, specifically GCN, GAT, and GIN. However, other types of GNNs, such as GraphSAGE and diffusion-based models like Graph Diffusion Networks (GDNs) and Diffusion Convolutional Neural Networks (DCNN), are also susceptible to oversmoothing. Testing GraphNormv2 on a broader set of GNN architectures would provide a more comprehensive evaluation and further validate its effectiveness across diverse model types.**
>
> Thank you for the comment. In line with the reviewer’s suggestion, we have conducted additional node classification experiments on Cora using GraphSAGE as the backbone under the same evaluation setup specified in the paper. The results are shown in the following table:
>
> |            | Node Classification | Node Classification (#layers=20) |
> |------------|---------------------|--------------------------------------|
> |            | Cora                | Cora                                 |
> | no norm    | $85.0 \pm 0.7 $     | $83.3 \pm 1.2 $                      |
> | batch      | $32.1 \pm 5.8 $     | $30.1 \pm 3.8 $                      |
> | graph      | $30.9 \pm 4.7 $     | $30.1 \pm 3.8 $                      |
> | pair       | $32.7 \pm 4.3 $     | $29.7 \pm 5.2 $                      |
> | group      | $\mathbf{87.2 \pm 0.7}$     | $85.6 \pm 0.9 $                      |
> | powerembed | $86.3 \pm 0.8$     | $85.2 \pm 0.6 $                      |
> | graph2 | $85.1 \pm 0.9 $     | $\mathbf{85.9 \pm 0.8} $                      |
>
>
> Our method, GraphNormv2 still performs competitively with baseline methods and outperforms the baselines in the deep architecture regime. Note that the message-passing operator for GraphSAGE is a row-stochastic matrix, hence its dominant eigenvector is the all-ones vector. We suspect that this might be the case why BatchNorm and GraphNorm perform so poorly with this backbone, as their centering step uses the all-ones vector and thus distorts the graph signal in the way as discussed in Section 5 of the paper.
>
>
> ----
> We appreciate your questions and comments very much. Please let us know for any further questions.

---

> > ### Comment · Reviewer_JBkc · 2024-11-23
> >
> > I thank the authors for their thorough clarifications. I have no further questions and will maintain my rating of 6. While heterophilic graph datasets are provided, they are simple and small, and the experiments are not sufficiently comprehensive to fully illustrate the theoretical results. Additionally, a score of 7 is not available.

---

> > > ### Author Response · Authors · 2024-11-27
> > >
> > > Thank you for your reply! We are glad that the rebuttal has answered your questions and that you considered raising your score to 7. We will try to give a more comprehensive evaluation including heterogeneous graphs in the camera ready version.

---

### Official Review · Reviewer_Kctt · 2024-11-01

**Soundness:** 3
**Presentation:** 4
**Contribution:** 3
**Rating:** 8
**Confidence:** 3

**Summary:**

Residual connections and normalization layers are commonly used in GNN models in order to hinder the oversmoothing phenomenon. Although successfully supported by empirical results, no theoretical investigations were undergone to explain this improvement. The authors of this paper bridge this gap and provide a better understanding as to why these heuristics are effective against oversmoothing.

Furthermore, backed by their findings, they propose a novel normalization layer that circumvents the highlighted limitation of existing approaches, namely the alteration of the information carried by the graph signal upon normalizing. Finally, extensive experiments are conducted, in support of their findings.

**Strengths:**

The presentation is very clear, with a nice balance of theoretical investigations coupled with empirical evaluations. The exposure is gradual, with relevant connections to existing literature. I really enjoyed reading this paper.

The technical results are detailed and extensive.

The authors conduct experiments on various real-world datasets, and the presented results display convincing performance.

**Weaknesses:**

The theoretical investigations consider the simplified setting of linearized GNNs. As mentioned in the paper, I acknowledge that you already consider bridging this gap in future work. Would it be possible to provide some insights into what specific challenges do you anticipate in extending the analysis to non-linear GNNs ?

You show that residual connections and normalization layers help against oversmoothing through different mechanisms. Do you think it would it be possible to quantify which of the two has the most significant impact against the oversmoothing phenomenon (be it through bounds or empirical measurement ) ?

**Questions:**

Please refer to the Weaknesses section.

**Details Of Ethics Concerns:**

Not applicable.

---

> ### Author Response · Authors · 2024-11-21
> **Response to Reviewer Kctt**
>
> We greatly appreciate your positive assessment and insightful comments. Below, we provide individual responses to your questions.
>
> **Q1: The theoretical investigations consider the simplified setting of linearized GNNs. As mentioned in the paper, I acknowledge that you already consider bridging this gap in future work. Would it be possible to provide some insights into what specific challenges do you anticipate in extending the analysis to non-linear GNNs?**
>
> Thank you for the good question.
>
> - Extending our analysis to fully nonlinear scenarios presents technical challenges. In our humble opinion, it would be difficult to develop a strong, unifying theory for non-convergence for all non-linearities used in practice. For example, a non-injective non-linearity (like ReLU) would be hard to handle when proving non-convergence due to theoretical edge cases. As a contrived counterexample, with ReLU activation, node features would oversmooth in a single step if the features are non-positive before applying the activation.
>
> - Moreover, many commonly used activation functions are contractions (such as ReLU, tanh or sigmoid), which is great for showing convergence, but would also present technical challenges for showing non-convergence. We do, however, believe that for BatchNorm, a convergence similar to Prop 4.7 is happening even with non-linear activations.
>
> **Q2: You show that residual connections and normalization layers help against oversmoothing through different mechanisms. Do you think it would it be possible to quantify which of the two has the most significant impact against the oversmoothing phenomenon?**
>
> Thank you for the question.
>
> - While we do not quantitatively compare how much these two methods alleviate overmoothing in this paper, our results reveal that both BatchNorm and residual connections face separate challenges in deep models. For BatchNorm, the convergence as in Prop 4.7 makes it such that it still “forgets” the initial features, although in a different way from the vanilla GNNs. For residual connections, while one can train a GNN with 100 layers using residual connections, such an architecture is not very likely to actually use all of these 100 layers, at least in the linear case. Let’s say $\alpha = 0.5$. Then, the features computed up to layer 90 only contribute $\approx 10^{-3}$ to the final features, where the last $10$ layers contribute the rest. The computation done in the first 70 layers is virtually inconceivable with a contribution of merely $10^{-9}$.  That means, while deep models with residual connections do train and also produce good results, these models might not be very parameter-efficient.
>
>
> - Moreover, we would like to also point out that quantifying how badly oversmoothing affects the system might not be directly correlated with performance gain. Oversmoothing is defined as a convergence to a one-dimensional space. Measuring “non-oversmoothing” would thus translate to measuring the “richness” of features. This could be done in a task agnostic manner, e.g. by analyzing the rank of the features - like we did in Figure 1 - or some other kind of measure of how different the features are from being rank 1. Yet these measures would not necessarily imply about the model task performance. For example, “some” amount of smoothing over the graph is beneficial for good performance in node classification task [1], as this is how message-passing uses the graph topology. To take this point to the extreme, a simple method to combat oversmoothing would be to add random noise to the features at each step. This would keep the features “rich”, but it is just as surely harmful to the performance. One could also think of measuring oversmoothing in a task-specific manner, e.g. measuring how nicely clusters in the final features $X^{(t)}$ correspond to the targets. This would be much more correlated with performance gain.
>
> We appreciate your questions and comments very much. Please let us know for any further questions.
>
> ----
> **References**
>
> [1] Wu et al. A non-asymptotic analysis of oversmoothing in graph neural networks. In ICLR 2023.

---

> > ### Comment · Reviewer_Kctt · 2024-11-22
> > **Response to authors**
> >
> > I thank the authors for the thoroughness of their clarifications. I do not have any further questions, and will keep my rating of 8 (accept) with pleasure.

---

> > > ### Author Response · Authors · 2024-11-22
> > > **Thank you!**
> > >
> > > We are glad to read this sentiment. We sincerely thank you for your valuable input!

---

### Official Review · Reviewer_fLzt · 2024-11-04

**Soundness:** 3
**Presentation:** 3
**Contribution:** 3
**Rating:** 6
**Confidence:** 3

**Summary:**

Residual connections and normalization layers are commonly used layers in deep learning.
The paper provides a theoretical analysis of the roles of residual connections and batch normalization in graph neural networks (GNNs) to mitigate the over-smoothing phenomenon.
Furthermore, it introduces a new normalization layer, GraphNormv2, which demonstrates enhanced performance in both graph classification and node classification tasks.

**Strengths:**

1). The writing and organization of this paper are very clear.

2). Provided theoretical support for the use of residual connections and normalization layers in GNNs.

**Weaknesses:**

1). The standard deviations for MUTAG, PROTEINS, PTC-MR, and Cora in Table 1 are quite large, making the experimental results less convincing, and why are the standard deviations for GIN so large? Why graphv2 does show significant improvement for GIN on the ogbn-arxiv dataset, but not for GCN and GAT?

2). Could you add a curve that shows the performance changes of different baselines and GraphNormv2 as the number of layers in GNNs increases?

3). There is an error in line 118.

**Questions:**

Refer to the content in the Weaknesses.

---

> ### Author Response · Authors · 2024-11-21
> **Response to Reviewer fLzt**
>
> We greatly appreciate your positive assessment of our work. Below, we provide individual responses to the comments you raised.
>
> **Q1: The standard deviations for MUTAG, PROTEINS, PTC-MR, and Cora in Table 1 are quite large, making the experimental results less convincing, and why are the standard deviations for GIN so large? Why graphv2 does show significant improvement for GIN on the ogbn-arxiv dataset, but not for GCN and GAT?**
>
> Thank you for the question.
>
> - We based our empirical evaluation on [1]. The standard deviations are in line with those reported in the paper. For example, on PROTEINS using GIN, they reported a standard deviation of $\pm 4.0$ while we report a standard deviation of $\pm 3.6$.
>
> - In the case of the GIN on ogbn-arxiv, we suspect that the issue for the baseline methods that GIN uses the unnormalized adjacency. This means that the degree of a node has a large impact on the magnitude of its feature vector. The scaling operation in normalization layers furthers the effect. As a result, Small degree nodes’ feature vectors are effectively pushed towards 0, making distinguishing them exceedingly hard. However, the centering operation in our GraphNormv2 using the dominant eigenvector counteracts this, as it will affect high degree nodes to a larger extent. We believe this is why our proposed method performs much better, while the other normalization methods seem to struggle. For GCN and GAT, the above phenomenon would not be so much of an issue, because the message-passing operators for these models are degree-normalized.
>
> **Q2:  Could you add a curve that shows the performance changes of different baselines and GraphNormv2 as the number of layers in GNNs increases?**
>
> In line with the reviewer's comment, we have added an experiment with a higher resolution in the depth of the GNNs. Please note that due to computation time constraints it was not possible to include all methods. We have thus selected the two best performing normalization methods (group and BatchNorm) alongside our proposed method. The results are presented in the following table. We have also added a preliminary figure to our appendix (see Appendix D.1), which we will aim to complete for camera ready submission.
>
> | depth | 5  | 10 | 15 | 20 | 25 | 30 |
> |--|----|----|----|----|----|----|
> graphv2 | 85.2 $\pm$ 0.7| 84.7 $\pm$ 1.0| 85.0 $\pm$ 0.6| 84.7 $\pm$ 0.9| 83.1 $\pm$ 1.5| 71.4 $\pm$ 2.3|
> batch | 85.0 $\pm$ 0.7 | 85.0 $\pm$ 0.7| 84.8 $\pm$ 0.7| 84.0 $\pm$ 1.0 | 82.0 $\pm$ 1.3| 65.1 $\pm$ 3.3|
> group |85.0 $\pm$ 1.1| 85.0 $\pm$ 0.8| 84.8 $\pm$ 2.1| 84.0 $\pm$ 0.9| 82.0 $\pm$ 1.6| 65.1 $\pm$ 2.3|
>
>
>
>
> **Q3: There is an error in line 118.**
>
> We apologize that we could not find the error you are referring to in line 118. Could you please elaborate? We would appreciate the chance to correct and improve.
>
> We appreciate your questions and comments very much. Please let us know for any further questions.
>
> ----
> **References**
>
> [1] Errica, Federico, et al. "A fair comparison of graph neural networks for graph classification." arXiv:1912.09893 (2019).

---

### Official Review · Reviewer_DmVD · 2024-11-05

**Soundness:** 3
**Presentation:** 3
**Contribution:** 3
**Rating:** 8
**Confidence:** 4

**Summary:**

This paper provides a theoretical foundation for understanding how residual connections and normalization layers affect oversmoothing in GNNs. It offers a detailed analysis of the mechanisms behind these techniques and proposes a new normalization layer that enhances the expressive power and performance of GNNs. The contributions are supported by both theoretical proofs and empirical evaluations.

**Strengths:**

1. It provides a thorough theoretical analysis of oversmoothing for residual connections and normalization layers.
2. The paper introduces a novel normalization layer that addresses a specific limitation of existing methods. The method is simple but effective. Random weights

**Weaknesses:**

1. The analysis is conducted only on linearized GNNs (except Remark 4.6) with random weights.
    a. Most GNNs are nonlinear, so we are more concerned with the properties of GNNs under nonlinearity. Can the analysis be generalized to nonlinear cases?
    b. Random weights usually appear during initialization. A network may have bad properties when it is initialized and good properties when it is trained, so we are more concerned about the trained GNNs which weights are not random.

2. The results for residual connections are not surprise. In equation (4), the update rule uses initial residual connections $\alpha X^{(0)}W_2^{(t)}$. In most cases ($W_2 ^ {(t)} $is not particularly bad), $X ^ {(t)} $contains $X ^ {(0)} $, will therefore not be collapsed. So the results are pretty obvious.

    For Proposition 4.2, if $x_i=0$, The conclusion is not valid. Please state the proposition strictly. For Proposition 4.3, I think it is smillar to Theorem 2 in GCNII [1], which is just a different statement. It is better to cite the relevant conclusions.

**Questions:**

1. From Figure 1, we can see that residual connection performs better than other normalizations (even graphnorm2).  So why use normalization layers? It seems that the residual connection is enough to prevent oversmoothing. GCNII can perform well under very deep architectures [1], but the performances of different normalization layers decrease.Results in Table 1 valide this.

2. Some results in Table 1 are strange. The performance of GIN with BatchNorm,PairNorm, GraphNorm, PowerEmbed and GroupNorm are not better than no norm on ogbn-arxiv, and worse than no norm very much for #layer=20 (6% v.s. 25.7). Is there any mistake? If the results are correct, this is confusing. Section 4.2 states that batch normalization is able to prevent collapse, but its performance is worse than no norm.

[1] Ming Chen, Zhewei Wei, Zengfeng Huang, Bolin Ding, and Yaliang Li. Simple and deep graph
convolutional networks. In ICML, 2020.

---

> ### Author Response · Authors · 2024-11-21
> **Response to Reviewer DmVD (1/2)**
>
> We appreciate your thoughtful comments and positive assessment of our work. After carefully reviewing your feedback, below we provide answers to the comments you raised.
>
> **Q1: The analysis is conducted only on linearized GNNs (except Remark 4.6) with random weights.**
>
> Thank you for the comment.
> - We agree with the reviewer, that the fundamental goal of this line of research is to understand the general case of GNNs, those that use nonlinearities. However, extending our analysis to fully nonlinear settings presents technical challenges.  For example, it would be hard to derive a similar theoretical statement with a non-injective nonlinearity (e.g. ReLU) as in the linearized cases, since one can construct contrived edge cases as counterexamples. As a result, many theoretical studies in GNN literature focus on linearized models for tractability [1,2,3], since they are more amenable to direct theoretical analysis and insights gained in the linear setting often carry over to more complex nonlinear GNNs.
> - Nonetheless, in our humble opinion, we believe that our work makes meaningful progress in understanding more complex nonlinear GNN architectures, as GNNs with normalization are nonlinear.
> We also agree that one should be both concerned with the initialized and the final trained networks for different reasons. In this work, we are taking a fundamental capability perspective: if oversmoothing is already present at initialization, there is simply not much the GNN can learn. In the architectures where we know that oversmoothing happens, it happens regardless of the weights (under mild assumptions, e.g. that the weights are not exponentially large). That is to say, any (trained or untrained) GNN will show the signs of oversmoothing and perform badly.
> - While our current analysis does not shed a light on the benefit of using BatchNorm or residual connections for the final, trained network, our results suggest deep GNNs with BatchNorm or residual connections, may be able to learn something useful, while it is simply not possible without these techniques.
>
> **Q2: The results for residual connections are not surprising. For Proposition 4.2, if $x_i=0$, The conclusion is not valid. Please state the proposition strictly. For Proposition 4.3, I think it is smillar to Theorem 2 in GCNII, which is just a different statement. It is better to cite the relevant conclusions.**
>
> Thank you for the comment.
>
> - Indeed, our theoretical findings confirm the observations of practitioners. Nevertheless, in order to conduct a comprehensive theoretical analysis of oversmoothing prevention, it is essential to incorporate both residual connections and normalization, as these are two principal methodologies for addressing this issue in the literature and in practice. There are also insights gained by comparing the two methods. In particular, our results suggest that these two methods achieve the effect through different mechanisms: residual connections alleviate oversmoothing by adding a portion of the initial signal, which thereby never decays, whereas normalization alleviates oversmoothing through the scaling operation, which lead to the node representations converging to the top-k eigenspace of the message-passing operator. These different mechanisms lead to different properties. For example, residual connections are able to “recover” rank in the unlikely event that the signal collapses - which BatchNorm cannot.
>
> - Thank you for your feedback for both Prop. 4.2 and 4.3. We would like to clarify that we stated in line 127/128 that $X_{:,j}^{(0)}$ should be normalized, which implies that it cannot be 0. We have added this requirement to 4.2 for better legibility.
>
> - Prop 4.3 indeed connects closely to Theorem 2 in [1]. To better clarify the distinction between the previous result and ours, please notice that our statement improves upon the previous result in the following ways:
>     - Theorem 2 in [1] implies that for any features in the Krylov subspace, there exists weights such that the corresponding GNN outputs such features. We prove in addition an upper bound, that the GNN cannot express any features that lie beyond the Krylov subspace.
>
>     - Our results hold for any GNNs in the form of (4) with residual connections, whereas Theorem 2 in [1] only accounts for GCNII.
>
> We have added the reference and discuss the above connection in the revised manuscript (Remark 4.4).

---

> > ### Author Response · Authors · 2024-11-21
> > **Response to Reviewer DmVD (2/2)**
> >
> > **Q3: From Figure 1, we can see that residual connection performs better than other normalizations (even graphnorm2). So why use normalization layers?**
> >
> > Thank you for the question. We would like to clarify that our claim in this paper is not that either method of preventing oversmoothing is superior, but the goal is to gain nuanced insights into their underlying mechanisms.
> >
> > - Figure 1 is for the purpose of empirically studying the convergence behavior of node features and cannot be directly used to draw conclusions on model performances. Although we can see in Figure 1 that residual connections are able to preserve a higher rank, it does not imply that model performance is necessarily better. To see this point, one could add random noise $\alpha Z$ ($Z_i,j \simeq \mathcal{N}(\mu, \sigma)$) instead of the initial signal $\alpha X^{(0)}$. This would also keep the features diverse and with high rank. However, it would not be useful for classification performance. For an example of the other extreme case, let’s assume that the binary target labels are $1$ if the node $i$ has an entry $v_i > c$ in the dominant eigenvector and $0$ otherwise. In this constructed case, converging to the rank 1 space of the dominant eigenvector will be beneficial for performance.
> >
> > - Our intuition why residual connections maintain a higher rank is that they make the state of features $X^{(t)}$ have much less effect on whether or not $X^{(t+1)}$ will become more smooth or not by creating a high rank feature at every step. Consider the extreme case that $X^{(t)}$ has converged to a matrix with numerical rank 2 (numerical rank meaning all singular values are $< \epsilon$). Then the probability for BatchNorm to produce a feature matrix $X^{(t+1)}$ with higher rank ($> 2$) is virtually 0. However, for residual connections, the proof of Prop 4.2 works even if $X^{(t)}$ has low rank as long as $X^{(0)}$ has high rank. Thus, w.h.p. $X^{(t+1)}$ will have a higher rank once more.
> >
> > **Q4: The performance of GIN with BatchNorm,PairNorm, GraphNorm, PowerEmbed and GroupNorm are not better than no norm on ogbn-arxiv, and worse than no norm very much for #layer=20 (6% v.s. 25.7). Section 4.2 states that batch normalization is able to prevent collapse, but its performance is worse than no norm.**
> >
> > Thank you for the comment. GIN did not train well in those cases. We suspect that the issue for the baseline methods is that GIN uses the unnormalized adjacency. This means that the degree of a node has a large impact on the magnitude of its feature vector. The scaling operation in normalization layers furthers the effect. As a result, small degree nodes’ feature vectors are effectively pushed towards 0, making distinguishing them exceedingly hard. This problem will compound after more iterations. However, the centering operation in our GraphNormv2 using the dominant eigenvector counteracts this, as it will affect high degree nodes to a larger extent. We believe this is why our proposed method performs much better, while the other normalization methods seem to struggle. For GCN and GAT, the above phenomenon would not be so much of an issue, because the message-passing operators for these models are degree-normalized.
> >
> > -----
> > We appreciate your questions and comments very much. Please let us know for any further questions.

---

### Author Response · Authors · 2024-11-27
**Manuscript Update**

We thank all reviewers for their thorough and helpful feedback. We are thrilled by the predominantly positive reception of our work. We have updated our manuscript to address questions and remarks:

- For legibility, we have added the condition that $|| X^{(0)}_{:,i} ||_2 = 1$ to the statement of Proposition 4.2.
- We have added a remark (Remark 4.4) to Proposition 4.3 that details the similarities and differences between the result established by [1].
- We have added additional experiments in appendix D.


We believe that these changes are beneficial to the overall readability of the paper, and we sincerely thank the reviewers for their invaluable contribution in enhancing our work.


[1]  Chen, Ming, et al. "Simple and deep graph convolutional networks." International conference on machine learning. PMLR, 2020.

---

### Meta-Review · Area_Chair_U9zZ · 2024-12-21

**Metareview:**

Residual connections and normalization layers have been introduced as effective empirical solutions to mitigate the oversmoothing problem. In the paper, the authors theoretically demonstrate the benefits of residual connections and normalization layers in linearized GNNs. They prove that (1) batch normalization can prevent a collapse of the output embedding space to a one-dimensional subspace and (2) the residual connection can prevent the signal from being too smooth.

All the reviewers agree that the theoretical contributions of the paper are novel, interesting, and valuable. The experiments are extensive and well-supported the theoretical findings in the paper. The presentation and writing of the paper are also clear and easy to follow. After the rebuttal, most of the concerns raised by the reviewers were addressed.

While there are some concerns about the theories are only established under the linearized settings of GNNs, I believe that the current theoretical contribution and findings of the paper are sufficiently novel for ICLR. As a consequence, I recommend accepting the paper. The authors are encouraged to incorporate the reviews and suggestions of the reviewers into the revision of their manuscript.

**Additional Comments On Reviewer Discussion:**

Please refer to the metareview.

---

### Decision · Program_Chairs · 2025-01-22

Accept (Poster)